# EVOLUTIONARY CACHING TO ACCELERATE YOUR OFF-THE-SHELF DIFFUSION MODEL

**Anirud Aggarwal   Abhinav Shrivastava   Matthew Gwilliam**
University of Maryland, College Park
`{anirud, mgwillia, abhinav2}@umd.edu`

## ABSTRACT

Diffusion-based image generation models excel at producing high-quality synthetic content, but suffer from slow and computationally expensive inference. Prior work has attempted to mitigate this by caching and reusing features within diffusion transformers across inference steps. These methods, however, often rely on rigid heuristics that result in limited acceleration or poor generalization across architectures. We propose **E**volutionary **C**aching to **A**ccelerate **D**iffusion models (ECAD), a genetic algorithm that learns efficient, per-model, caching schedules forming a Pareto frontier, using only a small set of calibration prompts. ECAD requires no modifications to network parameters or reference images. It offers significant inference speedups, enables fine-grained control over the quality-latency trade-off, and adapts seamlessly to different diffusion models. Notably, ECAD's learned schedules can generalize effectively to resolutions and model variants not seen during calibration. We evaluate ECAD on PixArt-$\alpha$, PixArt-$\Sigma$, and FLUX.1-dev using multiple metrics (FID, CLIP, Image Reward) across diverse benchmarks (COCO, MJHQ-30k, PartiPrompts), demonstrating consistent improvements over previous approaches. On PixArt-$\alpha$, ECAD identifies a schedule that outperforms the previous state-of-the-art method by 4.47 COCO FID while increasing inference speedup from 2.35x to 2.58x. Our results establish ECAD as a scalable and generalizable approach for accelerating diffusion inference. Our project page and code are available here: `https://research.aniaggarwal.com/ecad`

## 1 INTRODUCTION

Diffusion has emerged as the backbone for state-of-the-art image and video synthesis techniques (Dhariwal & Nichol, 2021; Ho et al., 2020; 2022; Liu et al., 2024c). Unlike prior methods involving deep learning, which would train a neural network to generate images in a single forward inference step, diffusion instead involves iterating over a prediction for many (20 to 50) steps (Lu et al., 2023). This process is quite expensive, and many researchers and practitioners try to reduce the latency while preserving, or even improving, the quality (Ma et al., 2023; Wimbauer et al., 2024; Selvaraju et al., 2024; Meng et al., 2023; Sauer et al., 2023). Some of these strategies involve training some model that can perform the inference in 1 to 4 steps, particularly with model distillation (Hinton et al., 2015; Meng et al., 2023). Other strategies do not train or tune any neural network weights, principally caching, where the diffusion model's internal features are reused across steps, allowing that computation to be skipped (Ma et al., 2023; Wimbauer et al., 2024; Li et al., 2023a).

We introduce a new conceptual and algorithmic framework for diffusion caching by reframing the problem and replacing existing heuristic-based approaches with a principled, optimization-driven methodology that is generalizable across model architectures. Existing caching methods typically offer a few discrete schedules, each with fixed trade-offs–for example, a 2x speedup with moderate quality loss, and a 3x speedup with greater degradation–without support for intermediate or more aggressive configurations. However, real-world deployments often operate under variable latency or quality constraints, necessitating further flexibility. We instead formulate caching as a multi-objective optimization problem, aiming to discover a smooth Pareto frontier that reveals a wide spectrum of speed-quality trade-offs. We show our frontiers for FLUX.1-dev (Labs, 2024) in Figure 1.

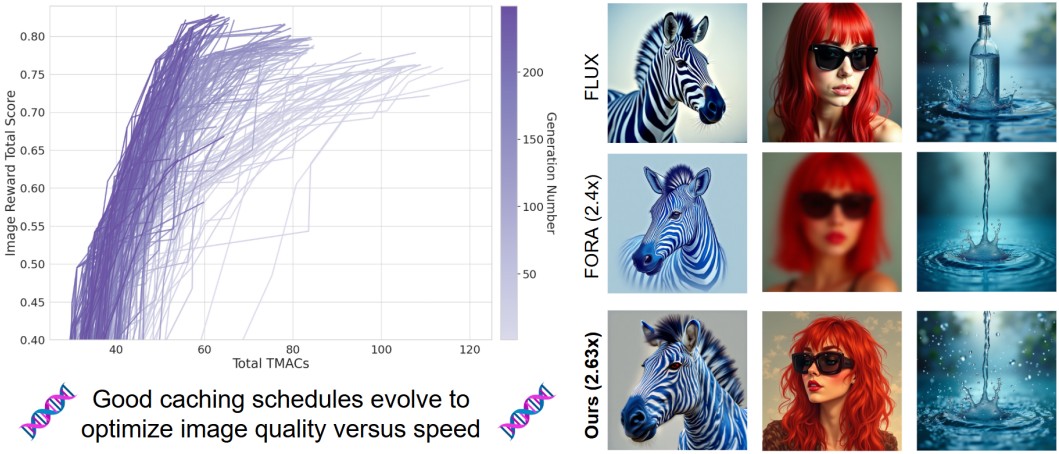

Figure 1: We conceptualize diffusion caching as a Pareto optimization problem over image quality and inference time and propose ECAD to discover such Pareto frontiers using a genetic algorithm. **Left**: performance progression over generations for FLUX.1-dev. **Right**: example 1024×1024 results with corresponding speedups.

Such frontiers are very challenging to produce given how caching schedules are currently derived. State-of-the-art approaches are motivated by heuristics, and key hyperparameters must be carefully hand-tuned by human practitioners based on performance on some set of key metrics (Selvaraju et al., 2024; Zou et al., 2025; Liu et al., 2024b; Zou et al., 2024; Liu et al., 2025a). We propose a different paradigm that does not rely on human-defined heuristics or hyperparameters, instead discovering effective caching schedules via genetic algorithm.

Our **E**volutionary **C**aching to **A**ccelerate **D**iffusion models (ECAD) requires two components: (i) some small set of text-only "calibration" prompts and (ii) some metric which computes image quality given a prompt and generated image–we use Image Reward (Xu et al., 2023a). We formulate caching schedules such that the genetic algorithm can automatically discover which features to cache (in terms of blocks and layer types) and when (which timestep). ECAD can be initialized with either random schedules or some set of promising schedules based on prior works such as Selvaraju et al. (2024); Liu et al. (2024b). Thus, while ECAD presents a different paradigm compared to prior works, it can also build on their valuable findings. ECAD takes these initial schedules and gradually evolves them according to the mating rules of a genetic algorithm, optimizing their "fitness" according to quality and computational complexity (measured in Multiply-Accumulate Operations, *aka* MACs).

This strategy is extremely flexible. While other methods are entirely designed around whether they cache entire block outputs, intermediate layer outputs (such as attention or feedforward outputs), or even specific tokens, ours is orthogonal to all of these. We offer a framework to optimize caching schedules according to any well-defined criteria. We instantiate it with schedule definitions and criteria in Section 3, but the general principles can be applied to arbitrary criteria and schedules to find Pareto-optimal caching frontiers. For example, fitness could be defined by human ratings of generated samples, and schedule definitions could be made more granular or coarse, or incorporate heuristics from other methods. Although our experiments target text-to-image synthesis, the framework is agnostic to modality and naturally extends to class-conditioned or text-to-video tasks. Furthermore, since ECAD computes no gradients and updates no weights, it introduces no memory overhead. During optimization, batch size is unrestricted (allowing use of single, small GPUs infeasible for distillation), and the process runs entirely asynchronously. Schedules could also be optimized for aggressively quantized models to further improve acceleration and quality.

Figure 1 showcases our method's strong performance and highlights flexibility across resolutions. Although optimized for FLUX.1-dev at 256×256, the same schedule applied to 1024×1024 still outperforms SOTA methods in both speed and quality. At 256×256, ECAD matches or surpasses un-accelerated PixArt-α and FLUX.1-dev baselines with 1.97x and 2.58x latency reductions, respectively. At more aggressive 2.58x and 3.37x settings, quality slightly drops but remains competitive.

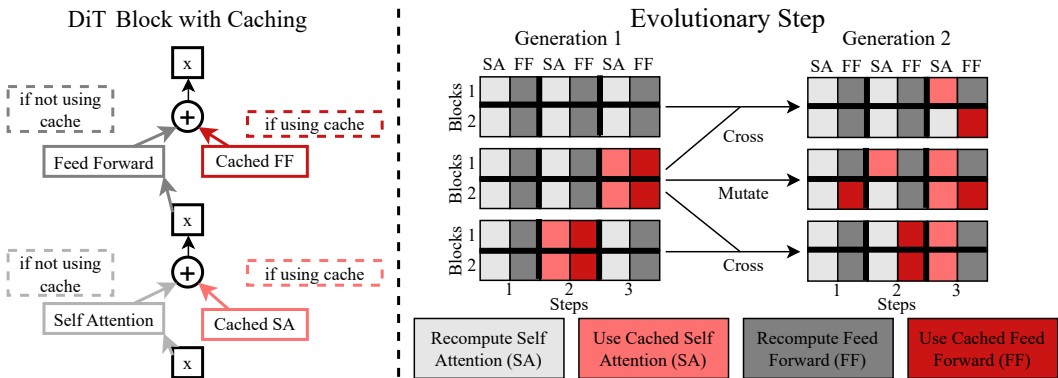

Figure 2: In the context of a transformer-based diffusion model, we describe how the transformer architecture allows for caching of attention and feedforward results separately **(left)**. We then give a toy illustration of how our method might transition from one generation to the next, prioritizing mating for schedules with the best quality-speed trade-offs **(right)**.

## 2 RELATED WORK

### 2.1 DIFFUSION FOR IMAGE AND VIDEO SYNTHESIS

Diffusion models predict noise, given noised image inputs, to generate high-quality images (Ho et al., 2020; Nichol & Dhariwal, 2021; Dhariwal & Nichol, 2021) and videos (Ho et al., 2022; Blattmann et al., 2023; Liu et al., 2024c). To save time and reduce feature sizes, these computations are typically performed in the latent space (Rombach et al., 2022) of a pre-trained variational autoencoder (Kingma & Welling, 2014). Although earlier works use U-Net backbones (Ronneberger et al., 2015), more recent methods rely mainly on transformer-based models (Vaswani et al., 2017; Dosovitskiy et al., 2020; Peebles & Xie, 2023; Bao et al., 2023), especially Diffusion Transformers (DiTs) (Peebles & Xie, 2023), which dominate the current landscape due to their powerful scaling properties (Chen et al., 2023; Esser et al., 2024; Liu et al., 2024c; Labs, 2024). Text-conditioning with multimodal models like CLIP (Radford et al., 2021), or extremely powerful text models like T5 (Raffel et al., 2023), allows for more granular control over image content (Saharia et al., 2022a; Ramesh et al., 2022; Nichol et al., 2022; Ruiz et al., 2023; Podell et al., 2023), not only in generative pipelines but also for editing (Kawar et al., 2023; Brooks et al., 2023; Sun et al., 2024; Ceylan et al., 2023; Chai et al., 2023).

### 2.2 ACCELERATING DIFFUSION INFERENCE

**Training.** Many works accelerate diffusion by training or fine-tuning models. Knowledge distillation (Hinton et al., 2015) trains a smaller or faster model to mimic the teacher, reducing steps but at high training cost and some quality loss (Salimans & Ho, 2022; Meng et al., 2023; Luo et al., 2023; Lee et al., 2024; Sauer et al., 2023; Kohler et al., 2024; Yin et al., 2023; Xu et al., 2023b). Other approaches train auxiliary modules to predict skip connections (Jiang et al., 2023), internal features (Gwilliam et al., 2025), caching configurations (Ma et al., 2024), or adaptive step schedules (Zhang et al., 2023). Network compression via pruning (Zhu et al., 2024; Fang et al., 2023) or quantization similarly requires retraining to recover accuracy, while post-training quantization offers limited gains in speed (Li et al., 2023b; Shang et al., 2023).

**Training-free.** An alternative direction accelerates inference without modifying model parameters by caching and reusing intermediate features. Early strategies designed for U-Nets (Li et al., 2023a; Ma et al., 2023; Wimbauer et al., 2024) do not transfer well to DiTs (Ma et al., 2024), which lack encoder–decoder hierarchy and rely only on within-block skip connections. Pioneering DiT caching works show promise, but some only cache entire blocks at fixed timestep intervals (Selvaraju et al., 2024), which sacrifices image quality, while others cache only attention layers (Liu et al., 2024b), which limits potential speedups. Recent works pursue finer-grained caching but depend heavily on

---

**Algorithm 1** Evolutionary Caching to Accelerate Diffusion models (ECAD)

---

**Require:** Diffusion model $M$, calibration prompts $P$, population size $n$, generations $G$, crossover probability $p_c$, mutation probability $p_m$

1: $\mathcal{P}_0 \leftarrow$ InitializePopulation($n$)         ▷ Random and heuristic-based schedules
2: **for** $g = 1$ to $G$ **do**
3:      **for** each schedule $S \in \mathcal{P}_{g-1}$ **do**
4:          $I \leftarrow M_S(P)$      ▷ Generate images $I$ using schedule $S$ on prompts $P$
5:          Compute quality metric $Q(P, I)$      ▷ Image Reward score
6:          Compute computational cost $C(S)$      ▷ MACs
7:      **end for**
8:      $\mathcal{P}_g \leftarrow$ Selection($\mathcal{P}_{g-1}$)      ▷ NSGA-II with Tournament Selection
9:      $\mathcal{P}_g \leftarrow$ Crossover($\mathcal{P}_g, p_c$)      ▷ Recombine schedules with 4-Point Crossover
10:      $\mathcal{P}_g \leftarrow$ Mutation($\mathcal{P}_g, p_m$)      ▷ Bit-flip mutation
11: **end for**
12: $\mathcal{F} \leftarrow$ ComputeParetoFrontier($\mathcal{P}_1, \mathcal{P}_2, ..., \mathcal{P}_G$)      ▷ Pareto frontier across all generations
13: **return** $\mathcal{F}$

---

heuristics and extensive hyperparameter tuning to balance efficiency and quality (Chen et al., 2024; Zou et al., 2025; Yuan et al., 2024; Liu et al., 2024a; 2025d; Qiu et al., 2025; Sun et al., 2025; Zou et al., 2024; Liu et al., 2025b;a; Bu et al., 2025). We build on these caching methods by replacing manual heuristic design and human-in-the-loop hyperparameter tuning with a genetic algorithm, leading to superior image quality.

## 3 METHODS

We begin by outlining key preliminaries for caching with DiTs (see Appendix A.1 for a general diffusion background). We then detail our method for modeling caching as a Pareto optimization problem over speed and quality, and the genetic algorithm used to optimize these frontiers.

### 3.1 PRELIMINARY: CACHING DIFFUSION TRANSFORMERS

DiTs utilize a modified transformer architecture optimized for the diffusion denoising process. A typical DiT block takes three inputs: a sequence of tokens $z'$ representing the noisy image, a conditioning vector $c$ (e.g., text embeddings), and a timestep embedding $t$. Caching in DiTs exploits temporal coherence between consecutive denoising steps. As the diffusion process proceeds from $z'_t$ to $z'_{t-1}$, the inputs to each block change gradually, creating an opportunity to reuse computed features from previous timesteps (Ma et al., 2023; Selvaraju et al., 2024). Rather than caching entire blocks, we employ component-level caching. For each transformer block, we selectively cache the outputs of specific functional components: self-attention ($f_{\text{SA}}$), cross-attention ($f_{\text{CA}}$), and feedforward networks ($f_{\text{FFN}}$). Formally, for a component $f_{\text{comp}}$ in block $b$ at timestep $t$, we can decide whether to compute it directly or reuse its cached value:

$$f_{\text{comp}}^b(z'_t, t, c) = \begin{cases} \text{compute}(z'_t, c, t) & \text{if recompute} \\ \text{cache}[f_{\text{comp}}^b, t+1] & \text{if cached} \end{cases}$$

When recomputing, the new value is stored in the cache for potential reuse in subsequent steps. Figure 2 demonstrates this for a DiT block with two components: self-attention and feedforward. The DiT's per-component residual connections allow features from the current inference step to be smoothly combined with cached features from previous steps.

This selective computation strategy can be represented as a binary tensor $S \in \{0, 1\}^{N \times B \times C}$, where $N$ is the number of diffusion steps, $B$ is the number of transformer blocks, and $C$ is the number of cacheable components per block. A value of 0 at position $(n, b, c)$ in $S$, which we show with shades of red in Figure 2, indicates that we reuse the cached value of component $c$ in block $b$ at diffusion step $n$ rather than recomputing it. A caching schedule directly impacts both computational efficiency and generation quality; aggressive caching (more 0s in $S$) reduces computation but may degrade output quality. Our method finds caching schedules with optimal trade-offs between computation and quality by identifying which components can be safely cached, in which blocks and timesteps.

Table 1: **Main results, 256×256, 20-step text-to-image generation.** We select schedules from our evolutionary Pareto frontier and compare them to prior works across various datasets and models on Image Reward, CLIP Score, and FID. Despite being optimized *only* on Image Reward with *just 100* calibration prompts, our method achieves superior results across other metrics and for unseen prompts.

| Model | Caching | Setting | TMACs↓ | ms / img↓ (speedup↑) | Image Reward↑ | Image Reward↑ | CLIP↑ | FID↓ | CLIP↑ | FID↓ | CLIP↑ |
|---|---|---|---|---|---|---|---|---|---|---|---|
| | | | | | Calibration | PartiPrompts | | MS-COCO2017-30K | | MJHQ-30K | |
| PixArt-α | None | | 5.71 | 165.74 (1.00x) | 0.90 | 0.97 | 32.01 | 24.84 | 31.29 | 9.75 | 32.77 |
| | TGATE | $m=15,k=1$ | 4.86 | 144.77 (1.14x) | 0.78 | 0.87 | 31.70 | 23.90 | 31.12 | 10.38 | 32.33 |
| | TGATE | $m=10,k=5$ | 3.47 | 108.52 (1.53x) | -0.051 | -0.27 | 28.90 | 29.78 | 28.29 | 17.52 | 29.38 |
| | FORA | $\mathcal{N}=2$ | 2.87 | 100.57 (1.65x) | 0.83 | 0.91 | **32.03** | 24.80 | 31.37 | 10.33 | 32.74 |
| | FORA | $\mathcal{N}=3$ | 2.02 | 82.55 (2.01x) | 0.60 | 0.83 | 31.94 | 24.50 | 31.35 | 11.11 | 32.63 |
| | ToCa | $\mathcal{N}=3,\mathcal{R}=60\%$ | 3.17* | 90.71 (1.83x)* | 0.71 | 0.76 | 31.46 | 22.05 | 30.99 | 12.01 | 32.37 |
| | ToCa | $\mathcal{N}=3,\mathcal{R}=90\%$ | 2.13* | 70.58 (2.35x)* | 0.60 | 0.68 | 31.35 | 24.01 | 30.92 | 11.80 | 32.35 |
| | DuCa | $\mathcal{N}=3,\mathcal{R}=60\%$ | 3.20 | 72.53 (2.29x)* | 0.76 | 0.79 | 31.53 | 23.13 | 31.03 | 11.69 | 32.48 |
| | DuCa | $\mathcal{N}=3,\mathcal{R}=90\%$ | 2.30 | 64.08 (2.59x)* | 0.76 | 0.74 | 31.42 | 24.69 | 30.96 | 12.53 | 32.39 |
| | **Ours** | fast | 2.13 | 84.09 (1.97x) | **0.96** | **0.99** | 31.94 | 20.58 | **31.40** | **8.02** | **32.78** |
| | **Ours** | faster | 1.46 | 69.17 (2.40x) | 0.90 | 0.88 | 31.44 | 21.93 | 31.10 | 9.92 | 32.34 |
| | **Ours** | fastest | **1.18** | **64.24 (2.58x)** | 0.81 | 0.77 | 31.53 | **19.54** | 31.28 | 8.67 | 32.24 |
| PixArt-Σ | None | | 5.71 | 167.62 (1.00x) | 0.85 | **1.08** | 31.90 | 24.63 | 31.11 | 10.53 | **32.65** |
| | FORA | $\mathcal{N}=3$ | 2.02 | 82.12 (2.04x) | 0.65 | 0.81 | 31.96 | 27.69 | 31.16 | 12.70 | 32.28 |
| | ToCa† | $\mathcal{N}=3,\mathcal{R}=60\%$ | 3.17* | 94.28 (1.78x)* | 0.11 | 0.19 | 31.03 | 54.80 | 30.34 | 35.42 | 30.64 |
| | ToCa† | $\mathcal{N}=3,\mathcal{R}=90\%$ | 2.13* | **73.03 (2.30x)*** | 0.07 | 0.14 | 30.89 | 56.48 | 30.25 | 36.53 | 30.55 |
| | **Ours** | fast | **1.91** | 84.84 (1.98x) | **0.85** | 1.02 | 31.86 | **22.17** | **31.25** | **8.91** | 32.52 |
| FLUX.1-dev | None | | 198.69 | 2620.09 (1.00x) | 0.69 | 1.04 | 31.88 | 25.76 | 30.95 | 17.77 | 31.06 |
| | FORA | $\mathcal{N}=3$ | 69.80 | 1073.70 (2.44x) | 0.67 | 0.93 | 31.88 | 23.51 | 31.30 | 19.38 | 31.10 |
| | ToCa | $\mathcal{N}=4,\mathcal{R}=90\%$ | **42.96*** | 1576.97 (1.66x)* | 0.63 | 0.93 | 31.81 | 23.78 | 31.26 | 21.59 | 30.88 |
| | DiCache | | 62.23 | 1161.86 (2.26x) | 0.61 | 0.97 | 31.97 | 26.18 | 31.12 | 20.70 | 31.18 |
| | TaylorSeer | $\mathcal{N}=5,\mathcal{O}=2$ | 59.88* | 1028.66 (2.55x)* | 0.29 | 0.54 | 31.16 | 29.66 | 30.19 | 24.36 | 30.64 |
| | TaylorSeer | $\mathcal{N}=6,\mathcal{O}=1$ | 49.97* | 865.97 (3.03x)* | -0.07 | 0.02 | 29.88 | 49.02 | 29.02 | 37.98 | 29.38 |
| | **Ours** | fast | 63.02 | 1016.59 (2.58x) | **0.83** | 1.04 | 32.24 | **21.61** | 31.58 | **16.14** | **31.69** |
| | **Ours** | fastest | 43.60 | **778.17 (3.37x)** | 0.69 | 0.89 | **32.27** | 26.66 | **31.63** | 21.43 | 31.67 |

†ToCa is not optimized for PixArt-Σ, so we reuse the hyperparameters from PixArt-α. Suboptimal results do not indicate that ToCa is not suitable for PixArt-Σ; instead, ToCa should be hand-optimized per-model.
*Refer to Appendix A.11 for a detailed explanation of MAC and latency calculations.

## 3.2 GENETIC ALGORITHM AS A PARADIGM FOR CACHING

**Caching, as Pareto frontiers.** The caching optimization problem inherently exhibits a trade-off between computational efficiency and generation quality. This can be formalized as a multi-objective optimization problem:

$$\min_{S}(C(S), Q(S))$$

where $C(S)$ denotes the computational cost function (lower is better) and $Q(S)$ represents the generation quality metric (lower is better, e.g., FID) for a caching schedule $S$. This optimization operates directly on the binary caching tensor $S \in \{0,1\}^{N \times B \times C}$ introduced previously. Possible configurations for $S$ naturally induce sets of solutions that form Pareto frontiers – improving one objective necessarily degrades the other. However, this search space is intractable to exhaustively explore, even for small DiTs, given current compute. Prior acceleration methods have predominantly relied on fixed heuristics that typically provide only isolated operating points. By contrast, our proposed approach explores a greater search space and discovers Pareto-optimal configurations, enabling practitioners to select schedules based on application-specific constraints.

**Evolutionary Caching to Accelerate Diffusion models (ECAD).** We introduce ECAD, an evolutionary algorithm-based framework for discovering efficient caching schedules for diffusion models, in Algorithm 1. Our approach's key insight is that the optimal caching configuration can be discovered through a population-based search over the space of possible caching schedules, using a small set of calibration prompts to evaluate candidate solutions. ECAD is a framework with 4 simple customizable components.

The practitioner may adjust granularity with the (1) **binary caching tensor shape** by adjusting $N$, $B$, and $C$ (the defaults we define for $S$ allow any component with a skip connection to be cached, on any block, for any timestep). While it does not require any image data, ECAD needs (2) **calibration prompts**, which we instantiate with the 100 prompts from the Image Reward Benchmark (Xu et al., 2023a). The practitioner can also select their preferred (3) **metrics**, where ideally both can be computed quickly online. We use Image Reward for quality, and MACs for speed (to avoid hardware

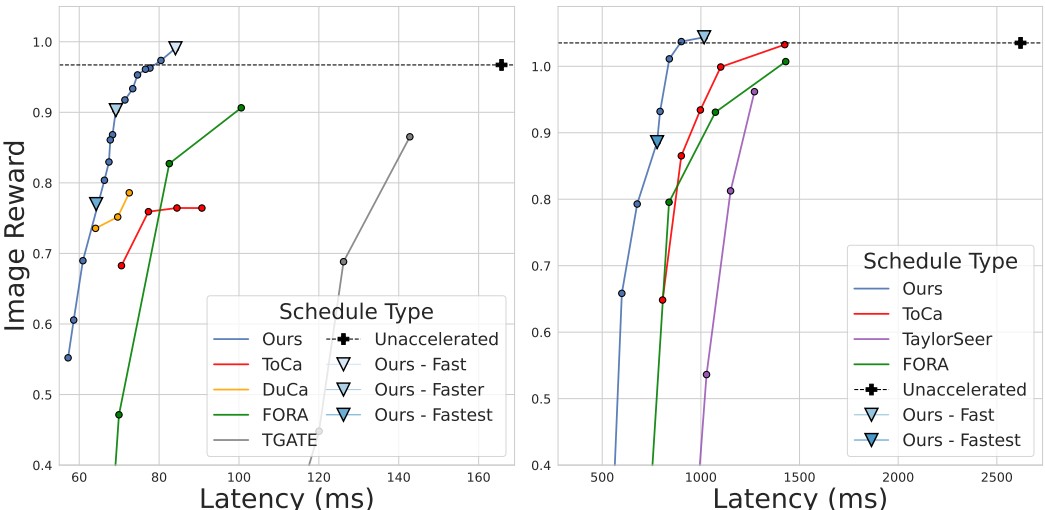

Figure 3: PartiPrompt Pareto frontiers at $256 \times 256$ for PixArt-$\alpha$ (**left**) and FLUX.1-dev (**right**).

dependencies). Then, we choose an (4) **initial population** of caching schedules, which should be diverse, and can be seeded based on prior knowledge (such as using FORA schedules) or initialized randomly. We utilize NSGA-II (Deb & Jain, 2013) for our genetic algorithm due to its efficient non-dominated sorting approach and proven effectiveness in multi-criteria optimization problems.

With all components defined, ECAD runs for the desired number of generations. In each generation, images are generated per caching tensor, and top-performing tensors (in quality and speed) evolve to form the next generation. This process incrementally improves Pareto frontiers for the selected model, scheduler, and timestep combination.

## 4    EXPERIMENTS

### 4.1    EXPERIMENTAL SETTINGS

**Model Architectures**   We provide experiments on three popular text-to-image DiT models: PixArt-$\alpha$, PixArt-$\Sigma$, FLUX.1-dev. Each model uses its default sampling method at 20 steps: DPM-Solver++ (Lu et al., 2023) for both PixArt models and FlowMatchEulerDiscreteScheduler (Esser et al., 2024) for FLUX.1-dev. Guidance scales are 4.5 for PixArt models and 5 for FLUX.1-dev. Both PixArt models employ 28 identical transformer blocks containing three components we enable caching for: self-attention, cross-attention, and feedforward. In contrast, FLUX.1-dev implements an MMDiT-based architecture (Esser et al., 2024) with 19 "full" and 38 "single" blocks. We enable caching for attention, feedforward, and feedforward context components in full blocks, and attention, MLP projection, and MLP output for single blocks. Cacheable component selection is discussed in Appendix A.2. We calibrate all models at $256 \times 256$ but evaluate at both $256 \times 256$ and $1024 \times 1024$.

**Evaluation Metrics**   We evaluate performance using Image Reward (Xu et al., 2023a), FID (Seitzer, 2020), and CLIP score (Zhengwentai, 2023) with ViT-B/32 (Dosovitskiy et al., 2020) on the Image Reward Benchmark prompts set (Xu et al., 2023a), the PartiPrompts set (Yu et al., 2022), MS-COCO2017-30K (Lin et al., 2015) (we use the same prompts and images as ToCa (Zou et al., 2025)) and MJHQ-30K (Li et al., 2024). On the Image Reward Benchmark prompts set, we generate each of 100 prompts at 10 different, fixed seeds for 1,000 total images. For PartiPrompts we generate a single image for each of the 1,632 prompts. To measure the speed of a particular caching schedule, we use two metrics: multiply-accumulate operations (MACs) and direct image generation latency. Except where otherwise stated, we utilize `calflops` (Ye, 2023) to measure MACs. We average end-to-end image generation latency using precomputed text embeddings on 1 NVIDIA A6000 GPU after discarding warmup runs; full details in Appendix A.11.

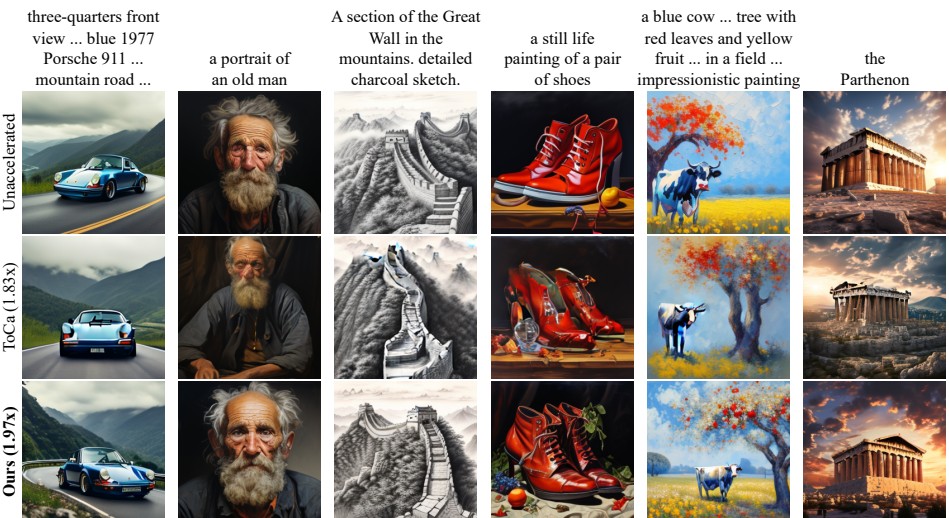

Figure 4: Qualitative results comparing our "fast" schedule for PixArt-$\alpha$ 256×256 with ToCa; see Figure 26 for FLUX.1-dev. "..." represent omitted text, see Appendix A.15 for full prompts.

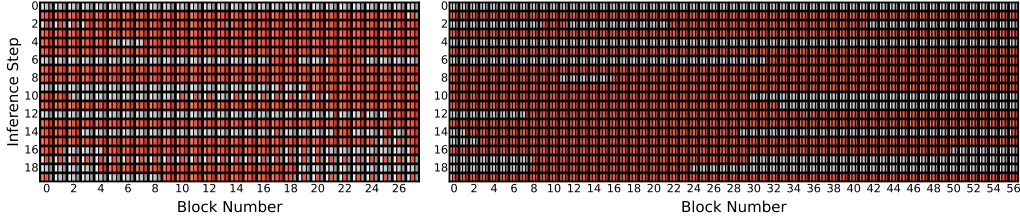

Figure 5: Our "fast" schedule for PixArt-$\alpha$ (**left**) and FLUX.1-dev (**right**). Reds are cached components and grays are recomputed (for PixArt-$\alpha$, from left to right: self-attention, cross-attention, and feedforward). See Appendix A.14 for more details.

## 4.2 MAIN RESULTS

We optimize ECAD on three diffusion models: PixArt-$\alpha$, PixArt-$\Sigma$, and FLUX.1-dev and present results for select schedules in Table 1. For PixArt-$\alpha$ at 256×256 resolution with 20 inference steps, we run 550 generations with 72 candidate schedules per generation, where each candidate generates 1,000 images (10 for each of 100 Image Reward Benchmark prompts). For FLUX.1-dev, we reduce the population to 24 schedules and evolve for 250 generations under otherwise identical settings. We initialize both using variants inspired by FORA and TGATE, detailed in Appendix A.8. For PixArt-$\Sigma$, we transfer 72 schedules from PixArt-$\alpha$'s 200th-generation Pareto frontier and run 50 additional generations, leveraging the models' shared DiT architecture.

Across all models, ECAD achieves strong performance on Image Reward (which correlates strongly with human preference (Xu et al., 2023a)) and FID. On PixArt-$\alpha$, our 'fastest' schedule reduces FID by 9.3 over baseline and by 2.51 over ToCa's best setting. On PixArt-$\Sigma$ and FLUX.1-dev, ECAD schedules outperform prior work and baseline by a significant margin. On FLUX.1-dev, our 'fast' schedule at 2.58x matches baseline Image Reward and the 'fastest' schedule at 3.37x maintains competitive quality. For prompt-image alignment, measured via CLIP score, ECAD roughly matches prior works, which is expected as caching should not affect prompt-image alignment.

We show full Pareto frontiers in Figure 3 on unseen prompts. ECAD discovers schedules that consistently outperform prior works across evaluation metrics while providing fine-grained control over the quality-latency trade-off. We provide qualitative comparisons for PixArt-$\alpha$ in Figure 4, with additional results in Figure 29 and further FLUX.1-dev comparisons in Figures 26, 27, 28. We show

Table 2: **Genetic scaling results.** We show performance changes as more iterations (generations) of ECAD run in terms of latency, PartiPrompts Image Reward, and MJHQ-30K FID. We select the schedule with highest TMACs per generation.

Table 3: **Model transfer results.** ECAD is first optimized on PixArt-$\alpha$ for 200 generations, and the resulting schedules are used to initialize optimization on PixArt-$\Sigma$ for an additional 50 generations (shown in the last row). Settings for both schedule discovery and evaluation are detailed below. We report TMACs, latency, Image Reward on the calibration and PartiPrompts set, and FID for MJHQ-30K. Transferring ECAD schedules between these two models results in only slight penalties to performance.

| # Gens | ms / img↓ (speedup↑) | Image Reward↑ | FID↓ |
|---|---|---|---|
| 1 | 145.09 (1.14x) | 1.00 | 9.40 |
| 50 | 92.76 (1.79x) | 0.98 | **7.97** |
| 150 | 87.11 (1.90x) | **1.00** | 8.11 |
| 300 | 86.62 (1.91x) | 0.99 | 8.04 |
| 500 | **76.52 (2.17x)** | 0.96 | 8.49 |

| Genetic Settings | | Evaluation Settings | | Latency | | Metrics | | |
|---|---|---|---|---|---|---|---|---|
| Model | Gens | Model | Res. | TMACs↓ | s / img↓ (speedup↑) | Calibration↑ | PartiPrompts↑ | FID↓ |
| PixArt-$\alpha$ | 200 | PixArt-$\alpha$ | 256 | 2.59 | **94.04 (1.76x)** | **0.96** | 1.02 | **8.00** |
| PixArt-$\alpha$ | 200 | PixArt-$\Sigma$ | 256 | 2.59 | 103.47 (1.62x) | 0.84 | **1.09** | 9.27 |
| PixArt-$\alpha$ | 250 | PixArt-$\alpha$ | 256 | 2.22 | 86.59 (1.91x) | **0.96** | 0.99 | **8.09** |
| PixArt-$\alpha$ | 250 | PixArt-$\Sigma$ | 256 | 2.22 | 93.68 (1.79x) | 0.79 | **1.06** | 9.06 |
| PixArt-$\Sigma$ | 50 | PixArt-$\Sigma$ | 256 | **1.91** | **84.84 (1.98x)** | 0.85 | 1.02 | 8.91 |

Table 4: **FLUX.1-dev detailed transfer results, $1024 \times 1024$ resolution, 20-step text-to-image generation.** We reuse our 'fast' schedule trained on FLUX.1-dev at 256x256 resolution, as well as an older, 'slow' schedule. We apply them for $1024 \times 1024$ image generation and compare them to prior works in terms of Image Reward, CLIP Score, and FID. Our results are competitive with prior work despite being evaluated at a different resolution than optimization.

| Model Settings | | Latency | | Calibration | PartiPrompts | | MS-COCO2017-30K | | MJHQ-30K | |
|---|---|---|---|---|---|---|---|---|---|---|
| Caching | Setting | TMACs↓ | s/img↓ (speedup↑) | Image Reward↑ | Image Reward↑ | CLIP↑ | FID↓ | CLIP↑ | FID↓ | CLIP↑ |
| None | | 1190.25 | 18.30 (1.00x) | 0.68 | **1.14** | 31.98 | 25.45 | 31.08 | **14.63** | 31.99 |
| None | 40% steps | 476.10 | 7.61 (2.41x) | 0.43 | 0.83 | 31.38 | 25.20 | 30.73 | 21.68 | 30.99 |
| FORA | $\mathcal{N} = 3$ | 416.88 | 7.62 (2.40x) | 0.27 | 0.69 | 31.20 | 29.45 | 30.52 | 24.65 | 30.69 |
| ToCa | $\mathcal{N} = 4, \mathcal{R} = 90\%$ | 300.41* | 7.42 (2.47x)* | 0.66 | 1.09 | 32.05 | 26.88 | **31.32** | 15.39 | 31.93 |
| TaylorSeer | $\mathcal{N} = 5, \mathcal{O} = 2$ | 357.39* | 7.20 (2.54x)* | 0.50 | 0.94 | **32.28** | 42.81 | 31.74 | 29.89 | 31.92 |
| **Ours** | slow$_{256 \to 1024}$ | 644.05 | 10.59 (1.73x) | **0.74** | 1.05 | 31.82 | **22.15** | 31.00 | 15.98 | 31.79 |
| **Ours** | fast$_{256 \to 1024}$ | 376.62 | **6.96 (2.63x)** | 0.71 | 1.05 | 31.88 | 26.69 | 30.91 | 17.76 | **31.99** |

the composition of the "fast" ECAD schedules for PixArt-$\alpha$ and FLUX.1-dev in Figure 5, with more schedules in Appendix A.14.

**Scaling Properties.** Unlike existing approaches, practitioners have the flexibility to run ECAD for as many generations as their time and compute constraints allow. While competitive schedules emerge within a few iterations, continued optimization yields steady improvements. To illustrate this, we track the 'slowest' schedule throughout the genetic process for PixArt-$\alpha$ and report results in Table 2. After just 50 generations, this schedule outperforms the unaccelerated baseline and all prior methods on Image Reward for unseen PartiPrompts and MJHQ FID. Further generations reduce latency at a slight cost in quality. Figure 6 shows the Pareto frontier for each generation on the calibration prompts; initial generations rapidly improve while later generations show incremental improvements.

## 4.3 EMERGENT GENERALIZATION CAPABILITIES

**Model Transfer Results.** To demonstrate ECAD's advantage over handcrafted heuristics, we transfer pre-optimized schedules between model variants. In Table 3, we select the "slowest" schedule from the Pareto-frontier across the first 200 generations of PixArt-$\alpha$ ECAD optimization and evaluate it on PixArt-$\Sigma$ as is, to demonstrate direct transfer results. Then, we perform an additional 50 optimization generations on PixArt-$\Sigma$ using 72 schedules transferred from the PixArt-$\alpha$ ECAD frontier at 200 generations. Although with direct transfer from PixArt-$\alpha$, PixArt-$\Sigma$ has higher latency than PixArt-$\alpha$ at 200 generations, after only 50 generations of optimization, it surpasses PixArt-$\alpha$'s speedup while improving calibration Image Reward and MJHQ FID. By comparison, simply transferring the 250-generation PixArt-$\alpha$ configuration yields only a 1.79x speedup instead of 1.98x, and has worse calibration Image Reward and MJHQ FID. This is a departure from recent caching innovations; for example, ToCa's carefully tuned PixArt-$\alpha$ settings cannot be transferred to PixArt-$\Sigma$ (see Table 1), despite the similarities between the two models.

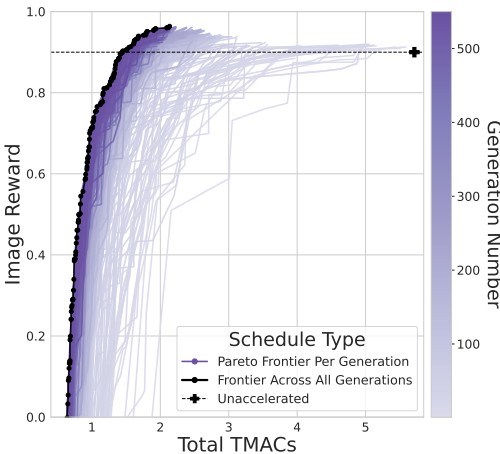

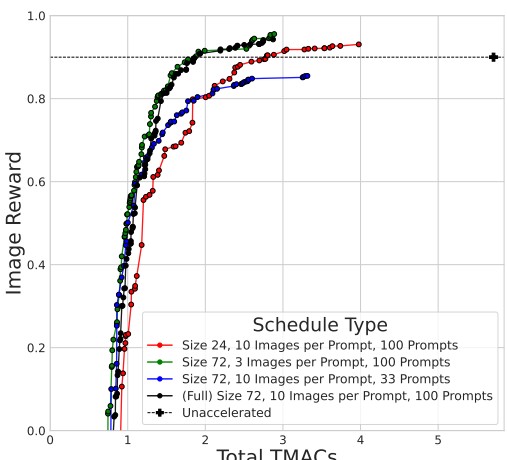

Figure 6: ECAD evolution. ECAD iteratively improves quality/time trade-offs as it evolves across generations as measured by Image Reward (PixArt-$\alpha$ 256$\times$256).

Figure 7: Faster ECAD optimization strategies. We compare "Full" ECAD to smaller population size, fewer images per prompt, and fewer prompts (PixArt-$\alpha$ 256$\times$256).

**Resolution Transfer Results.** We present ECAD's performance on FLUX.1-dev at 1024$\times$1024 resolution after optimization on 256$\times$256 in Table 4, and highlight its superior performance compared to FORA and the "None" approaches. We apply schedules as-is, with no further optimization at the higher resolution. While it is likely preferable to optimize ECAD at the target evaluation resolution if sufficient compute is available, we show this is not necessary in practice. In addition to the same 'fast' FLUX.1-dev schedule from Table 1 at 256$\times$256 resolution, we select a 'slow' schedule from just 50 generations of training at 256$\times$256. Despite ToCa being optimized for high resolution and ours for low resolution, our "fast" setting achieves superior Calibration Image Reward (a proxy for human preference) and COCO FID, and further surpasses concurrent TaylorSeer on unseen-prompt Image Reward, while avoiding its prohibitive memory overhead that reduced its batch size by 66%.

## 4.4 ABLATION ANALYSIS

To better explore the evolutionary algorithm's behavior, especially with respect to optimization time, we run three ablations with different hyperparameters on PixArt-$\alpha$ for 100 generations, varying the population size (from 72 to 24), the number of images generated per prompt (from 10 to 3), and the number of prompts used (from 100 to 33, selected randomly), each approximately reducing GPU time by 66%. The shape of the frontier of the reduced population setting in Figure 7 resembles previous generations of full populations settings, suggesting that reducing the population size is akin to running the model for fewer generations. Reducing the number of images per prompt is not notably harmful, while using a smaller set of only 33 prompts is very detrimental. However, as shown in Appendix A.4, this effect stems from size rather than diversity: smaller sets degrade quality, but equally sized sets with less diversity do not. Appendix A.4 further shows that a 100-prompt calibration set generated via ChatGPT performs comparably to the human-curated Image Reward set, demonstrating that large, diverse prompt collections are straightforward to assemble. In addition, we include ablations on NSGA-II hyperparameters in Appendix A.5, and display the effectiveness of alternative quality metrics in Appendix A.3.

## 5 DISCUSSION

**Limitations and Broader Impacts.** Optimizing on automatic metrics ties our performance to the quality of those metrics. We use Image Reward for the sake of cost and time; however, if we replace it with ranking by human users, for example, results could improve. ECAD does not introduce new societal risks beyond those inherent to diffusion models. While reduced inference cost may increase

potential for misuse, it also promotes broader image-generation accessibility and mitigates some environmental impact of image generation.

**Conclusion.** In this work, we reconceptualize diffusion caching as a Pareto optimization problem that enables fine-grained trade-offs between speed and quality. We provide a method, ECAD, which converts this problem into a search over binary masks, and can discover a best-case caching Pareto frontier. With only 100 text prompts, our method runs asynchronously with much lower memory requirements than training or fine-tuning a diffusion model. We achieve state-of-the-art results for training-free acceleration of diffusion models in both speed and quality.

ACKNOWLEDGMENTS

This work was partially supported by NSF CAREER Award (#2238769) to AS. The authors acknowledge UMD's supercomputing resources made available for conducting this research. The U.S. Government is authorized to reproduce and distribute reprints for Governmental purposes notwithstanding any copyright annotation thereon. The views and conclusions contained herein are those of the authors and should not be interpreted as necessarily representing the official policies or endorsements, either expressed or implied, of NSF or the U.S. Government.

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

# A APPENDIX

## A.1 DIFFUSION PRELIMINARY

Diffusion models have emerged as powerful generative models capable of producing high-quality images. In this section, we provide a brief overview of the diffusion process, the denoising objective, and the specific formulation for Diffusion Transformers (DiT).

**Basic Diffusion Process:** The diffusion process follows a Markov chain that gradually adds Gaussian noise to data. Given an image $x_0$ sampled from a data distribution $q(x_0)$, the forward diffusion process sequentially transforms the data into a standard Gaussian distribution through $T$ timesteps by adding noise according to a pre-defined schedule. This forward process can be formulated as:

$$q(x_t|x_{t-1}) = \mathcal{N}(x_t; \sqrt{1 - \beta_t}x_{t-1}, \beta_t\mathbf{I}) \tag{1}$$

where $\{\beta_t \in (0, 1)\}_{t=1}^{T}$ represents the noise schedule (Weng, 2021). We define $\alpha_t = 1 - \beta_t$ and $\bar{\alpha}_t = \prod_{s=1}^{t} \alpha_s$ for convenience. A key property arising from this process is that we can sample $x_t$ at any arbitrary timestep $t$ directly from $x_0$ without having to sample the intermediate states as:

$$x_t = \sqrt{\bar{\alpha}_t}x_0 + \sqrt{1 - \bar{\alpha}_t}\epsilon \tag{2}$$

where $\epsilon \sim \mathcal{N}(0, \mathbf{I})$. This property is particularly useful during training as it allows for efficient parallel sampling across different timesteps.

**Denoising Objective:** The denoising process aims to reverse the forward diffusion by learning to predict the noise added at each step. This is typically accomplished by training a neural network $\epsilon_\theta(x_t, t)$ to estimate the noise component in $x_t$. Its training objective is formulated as:

$$\mathcal{L} = \mathbb{E}_{t,x_0,\epsilon}[||\epsilon - \epsilon_\theta(x_t, t)||^2] \tag{3}$$

where $t$ is uniformly sampled from $\{1, 2, ..., T\}$, $x_0$ from the data distribution, and $\epsilon$ from $\mathcal{N}(0, \mathbf{I})$. During sampling, the noisy image is gradually denoised using various strategies. In the DDPM algorithm (Ho et al., 2020), the reverse process takes the form:

$$p_\theta(x_{t-1}|x_t) = \mathcal{N}(x_{t-1}; \mu_\theta(x_t, t), \sigma_t^2\mathbf{I}) \tag{4}$$

where $\mu_\theta(x_t, t) = \frac{1}{\sqrt{\alpha_t}}\left(x_t - \frac{\beta_t}{\sqrt{1-\bar{\alpha}_t}}\epsilon_\theta(x_t, t)\right)$. While effective, DDPM typically requires hundreds to thousands of denoising steps. For more efficient sampling, DPM-Solver++ (Lu et al., 2023) (used in both PixArt-$\alpha$ and PixArt-$\Sigma$) reformulates the diffusion process as an ordinary differential equation of the (simplified) form below:

$$\frac{dx}{dt} = -\frac{1}{2}\beta_t\nabla_x \log p_t(x) \tag{5}$$

DPM-Solver++ then applies high-order numerical methods to solve this ODE more efficiently. This leads to update rules that enable high-quality image generation in as few as 20 steps rather than the hundreds required by DDPM. However, each step still requires a forward pass through the noise prediction network, making the sampling process computationally intensive and a primary target for acceleration.

**DiT-specific Processing** Diffusion Transformers (DiT) adapt the transformer architecture for diffusion models, offering improved scalability compared to conventional UNet architectures. The processing pipeline for DiTs follows several key steps: first, the input image $x \in \mathbb{R}^{H \times W \times C}$ is encoded into a lower-dimensional latent representation using a pre-trained variational autoencoder (VAE): $z = \mathcal{E}(x) \in \mathbb{R}^{h \times w \times d}$, where $h$, $w$, and $d$ represent the height, width, and channel dimensions of the latent space, respectively. The latent representation is then divided into non-overlapping patches and linearly projected to form a sequence of tokens $z' = \text{Patch}(z) \in \mathbb{R}^{N \times d'}$, where $N = \frac{hw}{p^2}$ is the number of patches with patch size $p \times p$, and $d'$ is the embedding dimension of the transformer. Additionally, timestep embeddings and class or text condition embeddings are incorporated into the model to condition the generation process. Finally, the DiT model processes these tokens through a series of transformer blocks, each typically containing self-attention and cross-attention (or joint attention as in FLUX.1-dev), and feedforward network components.

Table 5: **Computation breakdown of a single transformer block forward pass at $256 \times 256$ resolution.** We report GMACs and each component's share of total block computation for PixArt-$\alpha$, PixArt-$\Sigma$, and FLUX.1-dev. Components marked as cache-enabled are those selected for caching in ECAD, as they dominate the computational cost. Components not selected are omitted for efficiency, not due to any fundamental limitation in their cacheability.

| Model | Component | Cache-Enabled | GMACs | % of Block Total |
|---|---|---|---|---|
| PixArt-$\alpha$ and PixArt-$\Sigma$ | Feedforward | Yes | 5.440 | 53.6 % |
| | Self-Attention | Yes | 2.720 | 26.8 % |
| | Cross-Attention | Yes | 2.000 | 19.7 % |
| | Ada Layer Norm Single | No | 0.000 | 0.0 % |
| | **Total: PixArt Transformer Block** | | **10.150** | **100 %** |
| FLUX.1-dev | Feedforward (Context) | Yes | 77.310 | 44.39 % |
| | Joint Attention (Mutli-stream) | Yes | 57.980 | 33.29 % |
| | Feedforward (Regular) | Yes | 38.650 | 22.19 % |
| | Ada Layer Norm Zero | No | 0.226 | 0.14 % |
| | Layer Norm | No | 0.000 | 0.00 % |
| | **Total: Flux Transformer Block, Full** | | **174.170** | **100 %** |
| | Linear (MLP Input Projection) | Yes | 72.480 | 41.66 % |
| | Linear (MLP Output Projection) | Yes | 57.980 | 33.32 % |
| | Joint Attention (Single-stream) | Yes | 43.490 | 24.99 % |
| | Ada Layer Norm Zero Single | No | 0.057 | 0.03 % |
| | GELU | No | 0.000 | 0.00 % |
| | **Total: Flux Transformer Block, Single** | | **174.000** | **100 %** |

Table 6: **Comparison of ECAD performance using Image Reward (IR) versus weighted sum of CLIP Score and CLIP Image Quality Assessment (IQA) as a quality metric.** The first 4 rows are duplicated from Table 1, while the final row displays results after running ECAD for 150 generations, where we optimize for TMACs and a weighted combination of CLIP Score (30%) and CLIP IQA. For CLIP IQA we specifically use Good (30%), Clean (20%), and Sharpness (20%) scores. CLIP Score encourages prompt-alignment, while CLIP IQA ensures the generated images are of high quality.

| Method | ms / img↓ (speedup↑) | TMACs | MJHQ FID↓ | COCO FID↓ | Calibration IR↑ | PartiPrompts IR↑ |
|---|---|---|---|---|---|---|
| PixArt-$\alpha$ Baseline | 164.74 (1.00x) | 5.71 | 9.75 | 24.84 | 0.90 | 0.97 |
| FORA $N = 2$ | 100.57 (1.65x) | 2.87 | 10.33 | 24.80 | *0.83* | *0.91* |
| ToCa $R = 60\%$ | *90.71 (1.83x)* | 3.17 | 12.01 | *22.05* | 0.71 | 0.76 |
| **Ours (IR)** | **84.09 (1.97x)** | **2.13** | **8.02** | **20.58** | **0.96** | **0.99** |
| **Ours (CLIP)** | 97.65 (1.68x) | *2.60* | *9.86* | 23.86 | 0.80 | 0.82 |

## A.2 CACHEABLE COMPONENT SELECTION

To enable ECAD on an off-the-shelf model, one must first select which components are cacheable. Any computation whose output can be stored at one step and reused at another–while introducing only minimal, acceptable inaccuracy–can be considered for caching. The number of such components determines the value of $C$ in the binary caching tensor $S \in \{0, 1\}^{N \times B \times C}$, introduced in Section 3. Since the search space grows linearly with $C$, careful selection is essential to ensure efficient and effective caching.

Note that the tensor notation is simplified for clarity. In cases where the model uses $k$ different types of DiT blocks, each with a different number of cacheable components, the caching tensor would instead take the form $S \in \{0, 1\}^{N \times (\sum_{i=1}^{k} B_i \times C_i)}$.

Table 5 enumerates the computational complexity of each DiT block's forward pass. We enable caching for the three most computationally expensive components per block, as they collectively dominate the total cost. Computations outside the DiT block's forward pass are not currently considered as they contribute less than 1% of the total compute.

A.3    QUALITY METRIC SELECTION

We select Image Reward as it is a strong indicator of human preference and is fast (Xu et al., 2023a). However, the ECAD framework supports a Bring-Your-Own-Reward paradigm, since image evaluation is done offline after each generation with image-prompt pairs. This makes human scoring easier than other online methods.

We include an ablation in Table 6 utilizing a weighted combination of CLIP Score and CLIP Image Quality Assessment (Wang et al., 2023), *aka* IQA, score to demonstrate the feasibility of alternative rewards. Since CLIP Score and CLIP IQA can be computed using the same CLIP image features, and we can precompute the text features for the calibration prompts offline, we still only need to perform a single ViT-L forward pass on the image to compute the metric. Thus, the cost is essentially the same as Image Reward. However, Image Reward is generally a superior metric to CLIP Score and CLIP IQA to judge image quality, so it is unsurprising that the results using it outperform CLIP variants.

Still, ECAD's robustness allows it to achieve good results, even with other metrics, which is a favorable property for application in more niche use cases. For example, one could utilize a human preference for video model, such as VisionReward (Xu et al., 2024), for text-to-video generation. Note that any number of metrics can be ensembled to dampen noisy reward signals (such as a combination of Image Reward, CLIP Score, and CLIP IQA).

*Generate 100 prompts for benchmarking image generation models (such as PixArt Alpha, Stable Diffusion, etc.). The prompts should be diverse and cover a wide range of styles, including photorealism, painting, anime, pixel art, and more. Each prompt should be crafted to evaluate aspects like aesthetics, compositional accuracy, text rendering, and subject diversity, in order to comprehensively test model quality.*

Figure 8: ChatGPT prompt used to generate a set of 100 diverse prompts.

*"Oil painting of rolling hills at sunrise, vibrant sky, wildflowers in foreground"*
*"Impressionist painting of a snowy mountain pass at twilight, soft pastels"*
*"Watercolor painting of a misty forest in autumn, golden leaves, tranquil stream"*
*"Classical landscape painting of a medieval village by a river, ornate details"*
*"Surrealist painting of a desert landscape with floating rocks and melting clocks"*
*"Abstract painting of a coastal landscape, bold shapes, bright primary colors"*

Figure 9: Sample of painting-style landscape prompts.

*Generate 100 prompts for benchmarking image generation models (such as PixArt Alpha, Stable Diffusion, etc.). The prompts should be brief (e.g. not granular), with no more than 5 words per prompt.*

Figure 10: ChatGPT prompt used to generate the set of 100 short (5-word) prompts.

*Generate 10 prompts for benchmarking image generation models (such as PixArt Alpha, Stable Diffusion, etc.). The prompts should be diverse and cover a wide range of styles, including photorealism, painting, anime, pixel art, and more. Each prompt should be crafted to evaluate aspects like aesthetics, compositional accuracy, text rendering, and subject diversity, in order to comprehensively test model quality.*

Figure 11: ChatGPT prompt used to generate the compact set of 10 diverse prompts.

## A.4 CALIBRATION PROMPT SELECTION

To demonstrate the ease of assembling a calibration prompt set and to isolate the factors that influence schedule quality, such as prompt source, domain specificity, and granularity, we evaluate ECAD across five distinct calibration strategies.

**Source and Curation:** We first compare the baseline *Image Reward Benchmark* (100 prompts) against two alternatives: *DrawBench200* (Saharia et al., 2022b), a human-curated set of 200 prompts with frequent repetitions, and a *GPT-Generated* set of 100 diverse prompts created via ChatGPT (see Figure 8). As shown in Table 7 and Figure 20, the schedule calibrated on the ChatGPT-generated set achieves nearly identical performance to the baseline. Interestingly, the Image Reward-calibrated schedule demonstrates superior generalization to DrawBench200 compared to the reverse scenario, suggesting that a smaller, more diverse set (Image Reward) provides a more robust foundation than a larger, repetitive one (DrawBench).

**Domain Specificity:** To test if the calibration domain restricts generalizability, we generate a set of 100 prompts exclusively describing *Painted Landscapes* (see Figure 9). Despite this narrow semantic focus, the resulting schedule maintains competitive performance on general-purpose benchmarks (COCO and MJHQ), performing similarly to the baseline. This indicates that ECAD relies less on semantic content matching and more on the complexity of the generation task itself.

**Quantity and Granularity:** Finally, we probe the limits of calibration data efficiency. We construct a *5-Word Prompts* set (100 coarse, short prompts; Figure 10) and a minimal *10 Prompts* set (see Figure 11. Table 8 reveals that prompt granularity is not a bottleneck; the schedule learned from 5-word prompts outperforms prior state-of-the-art methods with only a slight reduction in speedup. However, reducing the number of prompts to 10 causes noticeable degradation in MJHQ FID (increased to 10.02).

**Conclusion:** These results suggest that the *quantity* of prompts is the primary driver of schedule robustness, whereas the specific source, length, or semantic domain of the prompts is secondary. As such, gathering a sufficiently large (approx. 100) set of calibration prompts for ECAD is surprisingly simple and flexible.

Table 7: **Calibration prompt set source ablation.** Comparison of ECAD performance when calibrated on human-curated prompt sets Image Reward Benchmark and DrawBench200 versus a ChatGPT-generated set. Metrics include Image Reward (IR) on each prompt set, MJHQ-30K FID, CLIP score, and latency. Each result reflects the highest-TMACs schedule from the Pareto frontier after 100 generations. IRB, DB200, GPT-Gen, and PP refer to the Image Reward Benchmark, DrawBench200, GPT Generated, and PartiPrompts prompt sets, respectively.

| Calibration Prompt Set | # of calib. prompts | # imgs per prompt | ms / img↓ (speedup↑) | IRB IR↑ | DB200 IR↑ | GPT-Gen IR↑ | PP IR↑ | FID↓ | CLIP↑ |
|---|---|---|---|---|---|---|---|---|---|
| Image Reward Benchmark | 100 | 10 | 100.68 (1.65x) | **0.94** | 0.77 | 1.21 | **1.00** | 8.18 | 32.88 |
| DrawBench200 | 200 | 5 | **99.53 (1.67x)** | 0.87 | **0.79** | 1.19 | 1.00 | 8.90 | **32.93** |
| GPT Generated | 100 | 10 | 104.66 (1.58x) | 0.93 | 0.79 | **1.24** | 1.00 | **8.05** | 32.85 |

Table 8: **Calibration prompt diversity and size ablation.** We compare the baseline (Image Reward Benchmark) against domain-specific (Painted Landscapes), coarse (5-Word), and minimal (10 Prompts) calibration sets. Metrics include Image Reward (IR), MJHQ, and COCO. Each result reflects the highest-TMACs schedule from the Pareto frontier after 100 generations, except for "5 Word Prompts (Faster)", which is selected to provide an additional reference point for high-speed performance.

| Calibration Set | ms / img↓ (speedup↑) | Calib. Set IR↑ | PP IR↑ | MJHQ FID↓ | MJHQ CLIP↑ | COCO FID↓ | COCO CLIP↑ |
|---|---|---|---|---|---|---|---|
| ImageReward Benchmark | 100.68 (1.65x) | 0.94 | **1.00** | 8.18 | **32.88** | 21.40 | 31.48 |
| Painted Landscapes | 95.41 (1.74x) | 1.20 | 0.97 | 8.55 | 32.85 | 20.82 | **31.58** |
| 10 Prompts | 97.87 (1.69x) | **1.31** | 0.94 | 10.02 | 32.84 | 25.27 | 31.51 |
| 5 Word Prompts (Highest TMACs) | 109.18 (1.52x) | 1.01 | 0.98 | **7.42** | 32.82 | **20.05** | 31.46 |
| 5 Word Prompts (Faster) | **95.27 (1.74x)** | 1.00 | **1.00** | 8.76 | 32.76 | 21.68 | 31.36 |

A.5    HYPERPARAMETER ABLATIONS

To better characterize the behavior of ECAD, we conduct two ablations on two different sets of hyperparameters. The first is over the hyperparameters that are agnostic to the genetic algorithm used – population size, the number of images generated per prompt, and the number of prompts. The second is over the NSGA-II hyperparameters, but it should be noted that other genetic algorithms can be employed.

The former ablation is shown in Figure 7 and we include plots of the evolution over generations for each configuration. Figure 16 illustrates the impact of reducing the population size. This setting results in slightly noisier frontiers and slight performance degradation across all metrics: the MJHQ-30K FID worsens slightly and latency increases by 22 ms over the baseline–the largest increase among all ablations. Figure 17 examines the effect of reducing the number of images per prompt from 10 to 3, while keeping 100 prompts and a population of 72. This configuration achieves the fastest latency at 100.30 ms, the highest calibration Image Reward of 0.96, and the smallest increase in MJHQ-30K FID. In Figure 19, we reduce the number of prompts from 100 to 33 while maintaining 10 images per prompt. This setup exhibits the cleanest convergence behavior but significantly underperforms on calibration Image Reward and its final Pareto frontier is dominated by other settings. However, its PartiPrompts score remains competitive and it produces the best FID, suggesting the subset of prompts was challenging enough for some generalization. Detailed results for the highest-TMACs schedule after 100 generations under each hyperparameter setting are shown in Table 10.

The latter ablation, with results in Table 9, modifies one of each of the following hyperparameters: the number of crossover points, mutation probability, and whether direct copies are allowed. Refer to Appendix A.7 for the purpose of each of these. The results show disallowing direct copies of parents improves inference speed but significantly worsens FID ($8.60 \rightarrow 9.64$), as strong schedules are more frequently 'churned' with lower quality ones. Reducing the mutation rate to 1% has the greatest inference speedup, as it reduces exploration and increases exploitation, but results in poor quality. Conversely, reducing crossover to 1 point and increasing mutation rate to 15% both slow convergence. The high mutation rate promotes exploration and seems to prevent the high-FID local-minima seen in 1% mutation rate. Metrics across most configurations remain relatively stable, meaning a set of good-enough standard hyperparameters for the genetic algorithm is sufficient for ECAD.

Table 9: **Ablation comparing the effects of hyperparameters on NSGA-II**. Each row modifies exactly one hyperparameter, examining effects on computational cost (TMACs), latency, image quality metrics (Calibration Image Reward (IR), PartiPrompts IR), and MJHQ FID. All experiments use the highest-TMAC schedule after 100 generations, generating 3 images per prompt, with 100 prompts from the Image Reward prompt set, and with a population size of 72. Crossover probability ($P(\text{Cross})$) is the probability the parent's DNA is not directly copied to the offspring. $k$ Point crossover refers to the number of splices made to connect the parent's DNA, and the $P(\text{Mut})$ is the probability that an offspring will be mutated.

| Experiment Condition | TMACs | ms / img↓ (speedup↑) | Calibration IR↑ | PartiPrompts IR↑ | FID↓ |
|---|---|---|---|---|---|
| Baseline: $P(\text{Cross}) = 0.9$, 4 Point Cross, $P(\text{Mut}) = 0.05$) | 2.89 | 100.30 (1.65x) | 0.96 | 0.99 | 8.60 |
| No Direct Copies ($P(\text{cross} = 1.0)$) | 2.46 | 94.02 (1.76x) | 0.97 | 0.99 | 9.64 |
| 1 Point Crossover | 3.51 | 114.78 (1.44x) | 0.96 | 1.01 | 8.87 |
| 6 Point Crossover | 2.38 | 93.52 (1.77x) | **0.98** | **1.01** | **8.57** |
| $P(\text{Mutation}) = 0.01$ | **2.35** | **90.40 (1.83x)** | 0.97 | 0.98 | 9.11 |
| $P(\text{Mutation}) = 0.15$ | 3.97 | 127.83 (1.30x) | 0.96 | 1.00 | 8.70 |

Table 10: **Genetic hyperparameter ablation.** Performance of ECAD when varying population size, number of images per prompt, and number of calibration prompts. We report latency, Image Reward on calibration and unseen PartiPrompts, and MJHQ-30K FID. Each result corresponds to the highest-TMACs schedule lying on the Pareto frontier after 100 generations.

| Population Size | # imgs per prompt | # of calibration prompts | ms / img↓ (speedup↑) | Calibration IR↑ | PartiPrompts IR↑ | FID↓ |
|---|---|---|---|---|---|---|
| 72 | 10 | 100 | 100.68 (1.65x) | 0.94 | **1.00** | 8.18 |
| 24 | 10 | 100 | 122.96 (1.35x) | 0.93 | 1.00 | 8.92 |
| 72 | 3 | 100 | **100.30 (1.65x)** | **0.96** | 0.99 | 8.60 |
| 72 | 10 | 33 | 110.44 (1.50x) | 0.85 | 0.99 | **7.52** |

A.6 NUMBER OF INFERENCE STEPS ABLATION

We examine how the number of inference steps affects the performance of ECAD-generated schedules. Since ECAD produces schedules optimized for a particular step count, this ablation evaluates their robustness when applied at a different inference step setting.

We first learn ECAD schedules using 10 steps on DPM-Solver++ for 100 generations with standard hyperparameters, then upscale the binary mask of the schedule with the highest TMACs to 20 steps by duplicating each step. Formally, given a 10-step schedule $S_{10}$, we define step $i$ of the corresponding 20-step schedule $S_{20}$ as

$$S_{20}[i] = S_{10}\left[\left\lfloor \tfrac{i}{2} \right\rfloor\right], \quad i = 0, 1, \dots, 19$$

We similarly learn schedules at 20 steps and downscale to 10 steps by caching a component at step $i$ of $S_{10}$ only if it is cached in both corresponding steps $2i$ and $2i+1$ of $S_{20}$. Recalling that 0 indicates caching in $S$, for each block $b$ and component $c$ we define

$$S_{10}[i, b, c] = S_{20}[2i, b, c] \ \lor \ S_{20}[2i+1, b, c], \quad i = 0, 1, \dots, 9$$

Table 11 presents evaluation results. Applying a 10-step ECAD schedule at 20 inference steps yields improvements in both Image Reward and FID compared to the unaccelerated baseline. Conversely, using a conservative downscaling strategy reduces the overall speedup but still maintains performance gains over the baseline. Note that while the schedules here are evaluated *as-is*, they could also serve as starting points for further refinement or adaptation.

Table 11: **Inference step ablation.** We optimize ECAD on PixArt-$\alpha$ for 100 generations with standard hyperparameters, for 10 and 20 inference steps on DPM-Solver++. We then evaluate the highest-TMACs schedule from the Pareto frontier for both 10 and 20 steps, up- and down-scaling the learned caching schedules appropriately. Reported metrics include latency, Image Reward performance on the Image Reward prompt set (Calib. IR) and the unseen PartiPrompts set (PP IR), as well as both FID and CLIP scores on MJHQ-30K and MS-COCO2017-30K.

| Acceleration Type | Train Steps | Eval Steps | ms / img↓ (speedup↑) | Calib. IR↑ | PP IR↑ | MJHQ FID↓ | MJHQ CLIP↑ | COCO FID↓ | COCO CLIP↑ |
|---|---|---|---|---|---|---|---|---|---|
| None | 20 | 20 | 165.74 (1.00x) | 0.90 | 0.97 | 9.75 | 32.77 | 24.84 | 31.29 |
| ECAD | 20 | 20 | **100.68 (1.65x)** | **0.94** | 1.00 | 8.18 | 32.88 | 21.40 | 31.48 |
| ECAD | 10 | 20 | 121.04 (1.37x) | 0.94 | **1.01** | 8.80 | 32.74 | 21.67 | 31.33 |
| None | 10 | 10 | 89.85 (1.00x) | 0.84 | 0.90 | 10.83 | 32.77 | 25.82 | 31.42 |
| ECAD | 10 | 10 | **66.69 (1.35x)** | **0.93** | **0.97** | 8.35 | 32.62 | **22.02** | 31.40 |
| ECAD | 20 | 10 | 75.24 (1.19x) | 0.89 | 0.95 | 9.30 | **32.87** | 23.75 | **31.57** |

A.7 GENETIC ALGORITHM EVOLUTIONARY STEP IN DETAIL

The evolutionary step occurs once at the end of each generation to create new offspring for the subsequent generation. This step takes negligible time ($< 1$ minute) and does not require a GPU. Formally, this step can be understood as follows:

Given a population $P_g$ of size $n$ at generation $g$, ECAD employs the NSGA-II algorithm (Blank & Deb, 2020; Deb & Jain, 2013) to produce the next generation $P_{g+1}$ through the following steps:

1. **Selection and Offspring Generation:** An offspring population $Q_g$, also of size $n$, is generated from $P_g$ via binary tournament selection by repeating the following process until $Q_g$ is filled. Two pairs of candidates are randomly sampled from $P_g$. Within each pair, a tournament is conducted by first comparing candidates by Pareto rank, then breaking ties using crowding distance. The winners from each pair undergo crossover, followed by mutation, to generate offspring.

2. **Crossover:** With a probability of 0.9, we apply 4-point crossover to the binary caching tensors of the parent schedules. Four distinct crossover points are randomly selected along the flattened tensor, and two offspring are created by alternating segments between parents. With probability 0.1, the offspring are direct copies of their respective parents.

3. **Mutation:** Each candidate in $Q_g$ undergoes bit-flip mutation with a probability of 0.05. If selected, each bit in the binary tensor $S \in \{0, 1\}^{N \times B \times C}$ is independently flipped with

probability $\frac{1}{N \times B \times C}$. Note that after this step, we force all components in all blocks to be recomputed on the *first step*, since there is no 'cached' value to be reused.

4. **Non-Dominated Sorting:** The union $P_g \cup Q_g$ (size $2n$) is sorted into Pareto fronts $F_0, F_1, \ldots, F_d$ based on dominance. For each candidate $c$, we compute $\mathrm{Dom}_c(R)$, the number of candidates that dominate $c$ in some set of candidates $R$. Fronts are defined iteratively as:

$$F_0 := \{c \in P_g \cup Q_g \mid \mathrm{Dom}_c(P_g \cup Q_g) = 0\}$$
$$F_1 := \{c \in (P_g \cup Q_g) \setminus F_0 \mid \mathrm{Dom}_c((P_g \cup Q_g) \setminus F_0) = 0\}$$
$$\vdots$$
$$F_i := \{c \in (P_g \cup Q_g) \setminus \bigcup_{j=0}^{i-1} F_j \mid \mathrm{Dom}_c((P_g \cup Q_g) \setminus \bigcup_{j=0}^{i-1} F_j) = 0\}$$

Note that candidates in front $F_i$ are said to be of Pareto rank $i$; lower-rank candidates are 'fitter' solutions. Each front $F_i$ contains candidates not dominated by any candidate in fronts of higher rank.

5. **Population Selection:** The next generation $P_{g+1}$ is filled by sequentially adding complete fronts $F_0, F_1, \ldots$ until the population size $n$ is reached. If a front $F_k$ cannot be fully accommodated, it is sorted by crowding distance. The most diverse candidates–those with the fewest close neighbors–are selected to fill the remaining slots, always including the extrema to preserve frontier diversity.

## A.8 POPULATION INITIALIZATION

We initialize the first generation of schedules for PixArt-$\alpha$ using a diverse set of heuristic strategies informed by prior work. Each heuristic varies caching behavior based on step/block selection patterns:

- **Cross-Attention Only:** Cache cross-attention at $s$ evenly spaced steps. At each selected step, cache the cross-attention of $b$ DiT blocks, evenly spaced across the total 28 blocks.

- **Self-Attention Only:** Identical to the above, but cache only self-attention.

- **Feedforward Only:** Identical to the above, but cache only feedforward layers.

- **Cross- & Self-Attention, All Blocks:** Cache both cross- and self-attention for all blocks at every $n$th step.

- **FORA-inspired:** Following (Selvaraju et al., 2024), cache cross-attention, self-attention, and feedforward layers for all blocks at every $n$th step.

- **TGATE-inspired:** Following the gating mechanism from (Liu et al., 2024b), set gate step $m$ and interval $k$. After the first two warm-up steps, compute self-attention every $k$ steps, caching and reusing otherwise. After step $m$, self-attention is computed every step, while cross-attention is not recomputed and reuses the cached output from step $m$. Unlike TGATE, which averages the cross attention activation on text and null-text embeddings, we cache only the result from the text embedding.

The resulting Pareto frontiers for these heuristics are shown in Figure 12. From the complete set of generated schedules, we randomly select 72 to initialize ECAD's first generation for PixArt-$\alpha$.

For PixArt-$\Sigma$, as summarized in Section 4.2, we initialize with 72 schedules randomly sampled from the Pareto frontier of PixArt-$\alpha$ after 200 generations of ECAD optimization.

For FLUX.1-dev, we start with a FORA-inspired schedule, apply a few rounds of mutation and crossover, and randomly select 24 candidates to initialize ECAD.

When initializing populations, it is suggested to include at least one schedule that is nearly identical to the uncached baseline and one that is nearly fully-cached. The former will allow ECAD to find schedules with the highest image quality possible, and the latter will promote faster convergence to efficient schedules.

To better understand this, we analyze two random initialization strategies. We find that a naive random sampling of binary masks (*True Random*) is suboptimal; due to the Central Limit Theorem, candidate schedules cluster around the mean sparsity, failing to explore the extremes of the Pareto frontier (Figure 13).

To address this, we propose *Uniform Random Initialization*, which samples uniformly across the computational cost spectrum $[0, C_{max}]$. We first sample a target budget $C^* \sim \mathcal{U}(0, C_{max})$. We then determine valid integer counts $k_c$ for each component $c \in \{FF, SA, CA\}$ with GMAC cost $w_c$ by solving the linear Diophantine equation $\sum w_c k_c \approx C^*$. This is solved efficiently by iterating over the highest-weighted component ($k_{FF}$) and solving the remaining two-variable equation using the Extended Euclidean Algorithm:

$$w_{SA}k_{SA} + w_{CA}k_{CA} = C^* - w_{FF}k_{FF}$$

From the solution set, we sample a tuple $(k_{FF}, k_{SA}, k_{CA})$ and distribute the active flags uniformly across the $N \times B$ spatiotemporal positions.

As detailed in Table 12, while Heuristic initialization yields the best performance ($1.65\times$ speedup, 8.18 MJHQ FID), *Uniform Random* significantly outperforms *True Random* ($1.60\times$ vs. $1.28\times$ speedup) and prevents the population diversity collapse observed in the naive approach.

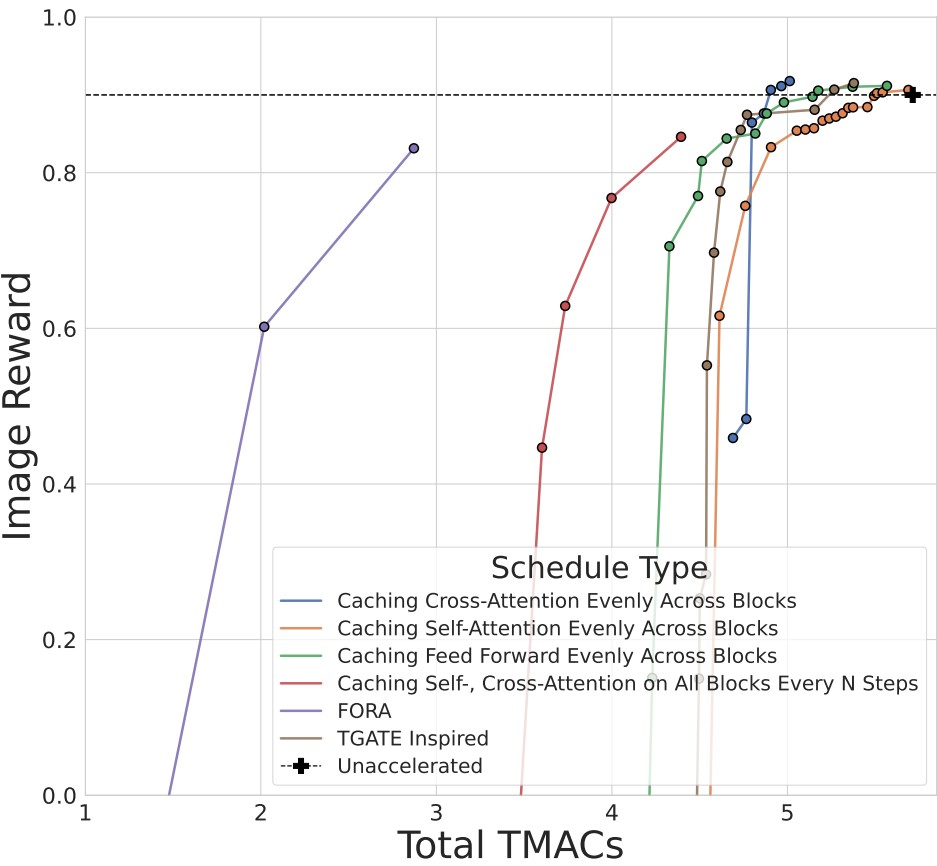

Figure 12: Pareto frontiers of Image Reward vs. computational cost for the handcrafted schedules described in Section A.8, evaluated on the Image Reward Benchmark. Notably, caching a single component (e.g., cross-attention or feedforward) offers slight gains over baseline. Among all heuristics, FORA achieves the best trade-off, with slightly lower quality but superior efficiency.

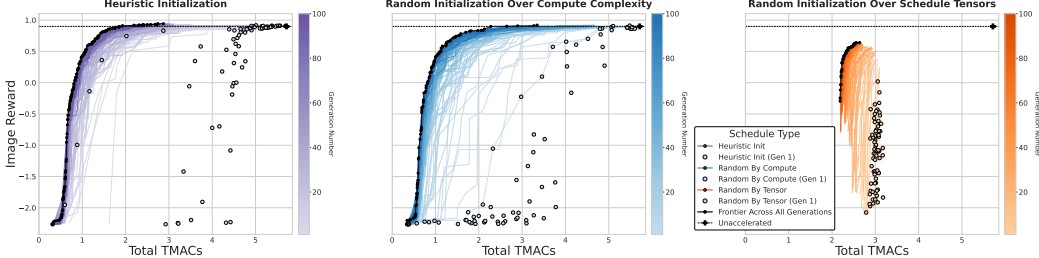

Figure 13: Schedule initialization Ablation. We initialize the first 72 candidates of generation 1 with three methods: heuristics as described in Section A.8, a random sample over the binary caching tensors, and sampling such that we have a uniform spread of compute complexity (TMACs). Heuristics converge the quickest, and achieve higher Image Reward performance. Uniform sampling over TMACs performs well, while randomly sampling caching schedule results in heavy grouping which prevents ECAD from optimizing effectively.

Table 12: **Initialization strategies ablation.** Comparison of ECAD performance using different initialization methods. Heuristic initialization yields the highest speedup and fidelity metrics compared to random initialization baselines. All evaluations are conducted on the schedule with the highest TMACs after 100 generations on default settings, except for 'Uniform Random (Closest)', which is selected to have the closest speedup to the Heuristic schedule for a more fair comparison.

| Initialization | ms / img↓ (speedup↑) | ImageReward IR↑ | PP IR↑ | MJHQ FID↓ | MJHQ CLIP↑ | COCO FID↓ | COCO CLIP↑ |
|---|---|---|---|---|---|---|---|
| Heuristic | **100.68 (1.65x)** | **0.94** | **1.00** | **8.18** | **32.88** | **21.40** | **31.48** |
| Uniform Random | 150.52 (1.10x) | 0.92 | 0.95 | 9.37 | 32.71 | 22.35 | 31.32 |
| Uniform Random (Closest) | 103.79 (1.60x) | 0.91 | 0.93 | 9.29 | 32.70 | 23.40 | 31.34 |
| True Random | 129.88 (1.28x) | 0.64 | 0.61 | 13.77 | 31.74 | 24.14 | 30.68 |

Table 13: **FLUX.1-dev performance on GenEval and DPG Bench.** We compare our and other methods from Table 1 on the GenEval and DPG Bench benchmarks using FLUX.1-dev (20 steps, $256 \times 256$). Our method does not impact GenEval Overall score at $2.58\times$ acceleration while other methods result in 2% to 22% quality decrease for lower acceleration. Our method achieves the highest speedups while even slightly improving the DPG Bench score, whereas other aggressive caching strategies degrade performance.

| Setting | | Latency | | GenEval Overall | | DPG Bench | |
|---|---|---|---|---|---|---|---|
| Caching | Setting | TMACs↓ | ms / img↓ (speedup↑) | Score | % Decrease | Score | % Decrease |
| None | | 198.69 | 2620.09 (1.00x) | 0.5842 | – | 22.7058 | – |
| ToCa | $N = 4, R = 90\%$ | **42.96**[*] | 1576.97 (1.66x)[*] | 0.5517 | 5.56% | 22.8215 | -0.51% |
| DiCache | | 62.23 | 1161.86 (2.26x) | 0.5699 | 2.45% | 22.6946 | 0.05% |
| TaylorSeer | $N = 5, O = 2$ | 59.88[*] | 1028.66 (2.55x)[*] | 0.4531 | 22.44% | 22.4695 | 1.04% |
| TaylorSeer | $N = 6, O = 1$ | 49.97[*] | 865.97 (3.03x)[*] | 0.3399 | 41.81% | 21.6869 | 4.49% |
| **Ours** | **Fast** | 63.02 | 1016.59 (2.58x) | **0.5892** | **-0.86%** | **22.8364** | **-0.58%** |
| **Ours** | **Fastest** | 43.60 | **778.17 (3.37x)** | 0.5258 | 10.00% | 23.5098 | -3.54% |

[*]Refer to Appendix A.11 for a detailed explanation of MAC and latency calculations.

## A.9 ADDITIONAL FLUX.1-DEV RESULTS

To further demonstrate the robustness of our method, we include supplementary quantitative results on the GenEval (Ghosh et al., 2023) and DPG Bench (Hu et al., 2024) in Table 13. Both benchmarks use the official prompt sets provided by each respective method, with 4 images generated per prompt. Our method's "Fast" schedule from Table 1 achieves $2.58\times$ acceleration with slightly higher performance on each metric as compared with the uncached baseline, while other methods result in quality degradation. Our "Fastest" schedule trades only some image quality to achieve $3.37\times$ acceleration.

## A.10 OPTIMIZATION COST AND LIMITATIONS OF ECAD

ECAD introduces an offline optimization phase that searches over binary caching schedules. This search is a one-time cost per model family: once a schedule (or set of schedules) is learned, it can be reused for that architecture and shared with downstream users, who simply choose an operating point on the quality–latency frontier.

In our PixArt configurations, the full "fast/faster/fastest" frontier requires $\approx$ 700 NVIDIA A6000 GPU-hours with a research-oriented implementation. However, competitive operating points can be obtained much more cheaply: the "fast" schedule with SOTA performance is discovered in only 358 generations (just 470 GPU-hours). 100 generations with default settings costs 145 GPU-hours, yielding a schedule with a 16% reduction in MJHQ FID over baseline. But with minor engineering changes, we achieve a schedule with an identical $1.65\times$ speedup, 11.8% MJHQ FID reduction over baseline, in *only 44 GPU-hours*. As such, these figures should be viewed as upper bounds given an under-optimized research framework.

The focus of this work is the algorithmic framework: formulating diffusion caching as a multi-objective optimization problem and demonstrating that a simple genetic algorithm can discover strong Pareto fronts across models and resolutions. System-level engineering—e.g., optimized kernels, greater hardware utilization, and `torch.compile` integration—is orthogonal to ECAD and can further reduce wall-clock search time without changing the method.

A key limitation is that ECAD adds an up-front compute cost. Nevertheless, unlike training-based accelerations, ECAD does not require gradients or weight updates, has lower VRAM requirements, and can be run asynchronously across heterogeneous, lower-end GPUs. For large-scale services employing ECAD, the one-time optimization cost is quickly amortized by per-sample latency savings.

Table 14: **Parameters used for latency evaluation.** $W$ is the number of warm-up batches discarded, $N$ is the number of batches used to compute the average latency, and $B$ is the largest batch size that fits in memory on a single NVIDIA A6000 GPU. All values are empirically chosen to ensure stable, consistent measurements and to match the quality benchmarking batch size.

| Model Name | Resolution | Warm-up ($W$) | Measured ($N$) | Batch Size ($B$) |
|---|---|---|---|---|
| PixArt-$\alpha$ | $256 \times 256$ | 1 | 5 | 100 |
| PixArt-$\Sigma$ | $256 \times 256$ | 1 | 5 | 100 |
| FLUX.1-dev | $256 \times 256$ | 1 | 10 | 18 |
| FLUX.1-dev | $1024 \times 1024$ | 5 | 25 | 3 |

## A.11 MAC AND LATENCY COMPUTATIONS

**Latency Setup:** Latency measurements are conducted on a single NVIDIA A6000 GPU for all models. For each model, we discard the first $W$ warm-up batches and compute the mean latency over the subsequent $N$ measured batches, using prompts from the Image Reward Benchmark. The reported per-image latency is obtained by dividing the average batch latency by the batch size $B$, except in the case of ToCa (see section below). Detailed configuration parameters are provided in Table 14.

**Latency Results:** The publicly available implementation for some prior works, denoted by $^*$ in tables, differed substantially from the infrastructure employed in our framework. While all methods use the same GPU (NVIDIA A6000) and identical warm-up and batch settings, ToCa and DuCa, for example, consistently produce higher latency measurements. To enable fair comparison, we normalize reported latencies by computing the relative speedup of each setting over its own baseline, then applying this speedup to our unaccelerated baseline latency:

$$\text{Normalized Latency}_{\text{Other}} = \frac{\text{Latency}_{\text{Other}}^{\text{cached}}}{\text{Latency}_{\text{Other}}^{\text{unaccelerated}}} \times \text{Latency}_{\text{Ours}}^{\text{unaccelerated}}$$

This procedure ensures that the reported values reflect performance improvements relative to each method's own baseline, enabling direct comparison across implementations. See Table 15 for details.

**ToCa MAC Results:** Multiply-accumulate operation (MAC) counts for ToCa are derived using the analytical formulations provided in the original work (Zou et al., 2025), specifically Section A.4. The relevant expressions are:

$$\text{MACs}_{SA} \approx 4N_1 D^2 + 2N_1^2 D + \frac{5}{2} N_1^2 H$$

$$\text{MACs}_{CA} \approx 2D^2(N_1 + N_2) + 2N_1 N_2 D + \frac{5}{2} N_1 N_2 H$$

$$\text{MACs}_{FFN} \approx 8N_1 D_{FFN}^2 + 12N_1 D_{FFN}$$

Here, $N_1$ and $N_2$ denote the number of image and text tokens respectively, $D$ is the hidden state dimensionality, $D_{FFN}$ refers to the dimensionality within the feedforward network, and $H$ is the number of attention heads. Results from DuCa (Zou et al., 2024), a concurrent method that builds upon ToCa, confirm that these approximations closely match empirical MAC counts.

**TaylorSeer MAC Results:** We compute MACs and FLOPs for all DiT models with the `calflops` from Ye (2023). However, when matching our configuration to that reported in Liu et al. (2025a), we find our computed FLOPs to always be different by a factor of exactly $1.249\times$ due to differences in implementation. As such, we report our computed values as is for consistency with other models, and note this scaling factor here.

Table 15: **Latency normalization details across different models and resolutions.** "True ms / img" refers to direct latency measured from the official implementation. "Speedup" is computed relative to each method's own unaccelerated baseline, and "Normalized ms / img" applies that speedup to our unaccelerated latency for fair comparison. Note we reduced batch size for TaylorSeer due to its high VRAM requirements.

| Model | Resolution | Implementation | Caching | Setting | True ms / img↓ | Speedup↑ | Normalized ms / img↓ |
|-------|-----------|----------------|---------|---------|---------------|----------|----------------------|
| PixArt-$\alpha$ | 256×256 | Ours | None | | 165.736 | | |
| | | ToCa | None | | 948.688 | 1.000x | 165.736 |
| | | ToCa | ToCa | $\mathcal{N}=3, \mathcal{R}=60\%$ | 519.258 | 1.827x | 90.715 |
| | | ToCa | ToCa | $\mathcal{N}=3, \mathcal{R}=90\%$ | 403.989 | 2.348x | 70.577 |
| | | DuCa | None | | 981.263 | 1.000x | 165.736 |
| | | DuCa | DuCa | $\mathcal{N}=3, \mathcal{R}=60\%$ | 429.405 | 2.285x | 72.527 |
| | | DuCa | DuCa | $\mathcal{N}=3, \mathcal{R}=90\%$ | 379.411 | 2.586x | 64.083 |
| PixArt-$\Sigma$ | 256×256 | Ours | None | | 167.624 | | |
| | | ToCa | None | | 925.024 | 1.000x | 167.624 |
| | | ToCa | ToCa | $\mathcal{N}=3, \mathcal{R}=60\%$ | 520.286 | 1.778x | 94.281 |
| | | ToCa | ToCa | $\mathcal{N}=3, \mathcal{R}=90\%$ | 403.038 | 2.295x | 73.035 |
| FLUX.1-dev | 256×256 | Ours | None | | 2620.095 | | |
| | | ToCa | None | | 3385.153 | 1.000x | 2620.095 |
| | | ToCa | ToCa | $\mathcal{N}=4, \mathcal{R}=90\%$ | 2037.433 | 1.661x | 1576.965 |
| | | ToCa | ToCa | $\mathcal{N}=5, \mathcal{R}=90\%$ | 1935.554 | 1.747x | 1499.949 |
| | | TaylorSeer | None | batch $=10$ | 2657.782 | 1.000x | |
| | | TaylorSeer | TaylorSeer | $\mathcal{N}=5, \mathcal{O}=2$ | 1043.457 | 2.547x | 1028.661 |
| | | TaylorSeer | None | batch $=18$ | 2630.581 | 1.000x | |
| | | TaylorSeer | TaylorSeer | $\mathcal{N}=6, \mathcal{O}=1$ | 869.438 | 3.026x | 865.972 |
| | 1024×1024 | Ours | None | | 18297.603 | | |
| | | ToCa | None | | 34109.719 | 1.000x | 18297.603 |
| | | ToCa | ToCa | $\mathcal{N}=4, \mathcal{R}=90\%$ | 13832.082 | 2.466x | 7419.995 |
| | | TaylorSeer | None | batch $=1$ | 18947.390 | 1.000x | |
| | | TaylorSeer | TaylorSeer | $\mathcal{N}=5, \mathcal{O}=2$ | 7452.669 | 2.542x | 7197.085 |
| | | TaylorSeer | TaylorSeer | $\mathcal{N}=6, \mathcal{O}=1$ | 6219.621 | 3.046x | 6006.323 |

## A.12 COMPARISON TO CONCURRENT WORKS

Although our method is thoroughly evaluated against established baselines, comparison with concurrent works is limited. As such, we restrict our comparisons to methods with publicly available code at the time of submission. We note several relevant differences nonetheless. First, concurrent methods such as SpeCa (Liu et al., 2025c) and ClusCa (Zheng et al., 2025) report high acceleration figures partly attributable to a 50-step inference setting in addition to their strong method. Our experiments use 20 steps, which is already approximately $2.5\times$ faster than 50-step inference with minimal quality

loss. Second, while ClusCa reduces memory overhead relative to TaylorSeer (Liu et al., 2025a), it still incurs roughly 10% additional cost (Zheng et al., 2025), which in practice constrains batch size. In contrast, ECAD introduces no memory overhead. Finally, because both SpeCa and ClusCa rely on manually tuned hyperparameters (e.g., propagation ratio, cluster size, cache interval), ECAD's optimization framework could be utilized to automatically discover such configurations.

## A.13 ADDITIONAL ECAD OPTIMIZATION PLOTS

Figure 14 illustrates the progression of ECAD optimization for PixArt-$\Sigma$ and FLUX.1-dev at $256 \times 256$ resolution. PixArt-$\Sigma$ converges rapidly, likely due to its initialization from pre-optimized schedules learned on PixArt-$\alpha$. FLUX.1-dev converges to a steeper Pareto frontier, with its resulting schedules substantially outperforming the unaccelerated baseline on the Image Reward benchmark. We hypothesize that this steep convergence is facilitated by an initial population with a relatively high mean acceleration. See Section A.8 for additional details on population initialization.

Additionally, we include the Pareto frontier of PixArt-$\Sigma$ as measured by Image Reward on the unseen PartiPrompts set vs. image generation latency in Figure 15. Our method achieves Pareto dominance over FORA but does not reach the unaccelerated baseline's level of performance.

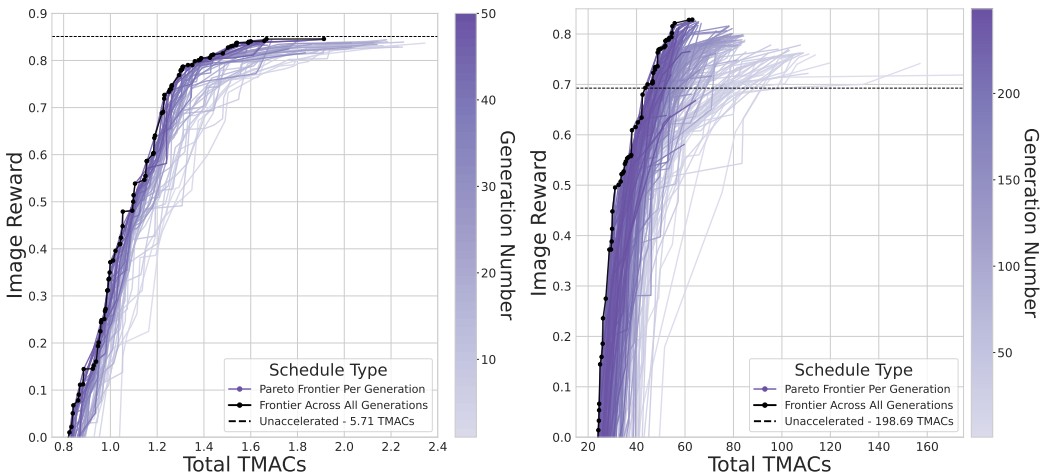

Figure 14: Progress of ECAD optimization as measured by Image Reward and TMACs. **Left**: PixArt-$\Sigma$ optimized for 50 generations, initialized using 200 generations of PixArt-$\alpha$ optimization. **Right**: FLUX.1-dev optimized for 250 generations, initialized using basic heuristics.

## A.14 VISUALIZING ECAD SCHEDULES

To better understand how ECAD optimizes caching schedules under different constraints and settings, we visualize selected schedules using heatmaps. Each heatmap represents a schedule, where red shades indicate cached components and gray shades indicate recomputed components. For PixArt models, the component order left-to-right is self-attention, cross-attention, and feedforward. FLUX.1-dev uses two types of DiT blocks. Block numbers 0 to 18 are 'full' FLUX DiT blocks, whose components are multi-stream joint-attention, feedforward, and feedforward context. Blocks 19 to 56 are 'single' blocks with components single-stream joint-attention, linear MLP input projection, and linear MLP output projection. Figure 18 and Figure 24 show representative schedules for PixArt-$\alpha$ and PixArt-$\Sigma$ used throughout the paper. Figure 22 compares FLUX.1-dev's 'slow' and 'fastest' schedules. Furthermore, Figure 23 visualizes how ECAD schedules evolve over time for PixArt-$\alpha$, comparing the highest-TMACs candidate at generations 50, 200, and 400. Finally, Figure 25 presents the highest-TMACs schedules resulting from our genetic hyperparameter ablations, illustrating how variations in population size impact the structure of learned caching strategies.

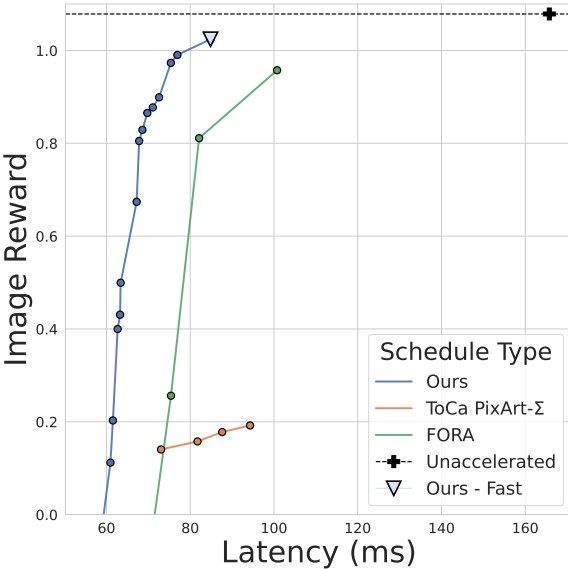

Figure 15: PartiPrompt Image Reward vs. latency for PixArt-$\Sigma$. Note that ToCa is not optimized for PixArt-$\Sigma$ and its parameters are transferred from PixArt-$\alpha$. Our method achieves Pareto dominance with a significant margin, but does not reach baseline performance.

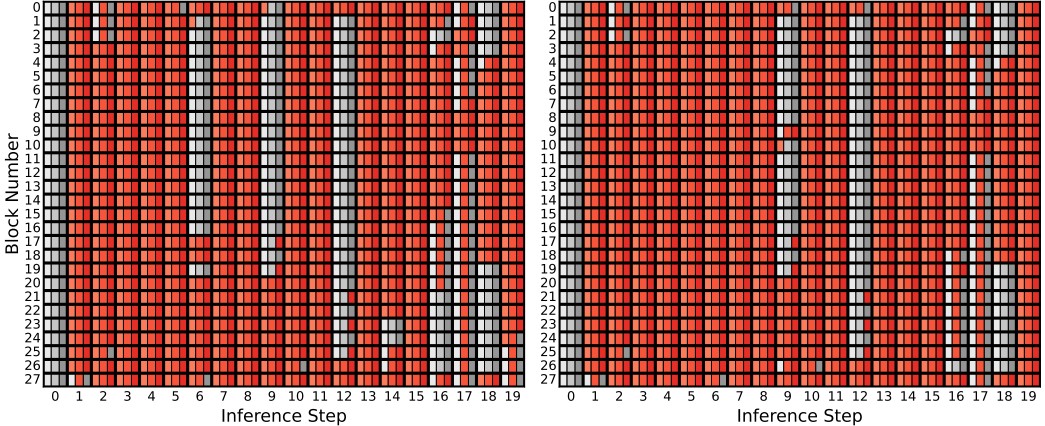

Figure 18: ECAD schedules for PixArt-$\alpha$ from Table 1: "faster" (**left**) and "fastest" (**right**). Despite being separate schedules with no guarantee of relation, the "faster" schedule has near identical structure to "fast", with more caching along steps 6 and 16. Furthermore, it appears cross-attention matters less than self-attention and the feedforward network during steps 16 and 17 and can safely be cached.

## A.15    FURTHER QUALITATIVE RESULTS

In addition to the PixArt-$\alpha$ $256\times256$ results shown in Figure 4, we present further qualitative comparisons using FLUX.1-dev at $256\times256$ (Figure 26) and $1024\times1024$ (Figure 27). Notably, in prompts such as "I want to supplement vitamin c, please help me paint related food," our method exhibits stronger prompt adherence than both the uncached baseline and ToCa. This behavior is likely influenced by ECAD's optimization for the Image Reward metric, which emphasizes semantic alignment with the prompt.

**Full Prompts from Figure 4**, from left to right:

- "Three-quarters front view of a blue 1977 Porsche 911 coming around a curve in a mountain road and looking over a green valley on a cloudy day."

- "a portrait of an old man"

- "A section of the Great Wall in the mountains. detailed charcoal sketch."

- "a still life painting of a pair of shoes"

- "a blue cow is standing next to a tree with red leaves and yellow fruit. the cow is standing in a field with white flowers. impressionistic painting"

- "the Parthenon"

**Full Prompts from Figure 26, 27**, from top-to-bottom:

- "Drone view of waves crashing against the rugged cliffs along Big Sur's Garay Point beach. The crashing blue waters create white-tipped waves, while the golden light of the setting sun illuminates the rocky shore."

- "Bright scene, aerial view, ancient city, fantasy, gorgeous light, mirror reflection, high detail, wide angle lens."

- "3d digital art of an adorable ghost, glowing within, holding a heart shaped pumpkin, Halloween, super cute, spooky haunted house background"

- "8k uhd A man looks up at the starry sky, lonely and ethereal, Minimalism, Chaotic composition Op Art"

- "I want to supplement vitamin c, please help me paint related food."

- "A deep forest clearing with a mirrored pond reflecting a galaxy-filled night sky."

- "A person standing on the desert, desert waves, gossip illustration, half red, half blue, abstract image of sand, clear style, trendy illustration, outdoor, top view, clear style, precision art, ultra high definition image"

**Full Prompts from Figure 28**, from top-to-bottom:

- "Eiffel Tower was Made up of more than 2 million translucent straws to look like a cloud, with the bell tower at the top of the building, Michel installed huge foam-making machines in the forest to blow huge amounts of unpredictable wet clouds in the building's classic architecture."

- "Mural Painted of Prince in Purple Rain on side of 5 story brick building next to zen garden vacant lot in the urban center district, rgb"

- "Editorial photoshoot of a old woman, high fashion 2000s fashion Steampunk makeup, in the style of vray tracing, colorful impasto, uhd image, indonesian art, fine feather details with bright red and yellow and green and pink and orange colours, intricate patterns and details, dark cyan and amber makeup. Rich colourful plumes. Victorian style."

**Full Prompts from Figure 29**, from top-to-bottom:

- "a handsome villain in his early 40s with very short bleach blonde hair and glowing red eyes wearing a blue armor and red cape. hyperrealistic, mythological, regal, 8k, medieval."

- "logo, simplistic, art style, multiple parallel universes together, different ages and themes over an open book "

- "professional Food photography, BeerenProteinSmoothie in a glass decorated with a mint leaf, high quality, hyper, detailed, beautifully color, beautifully color graded, cinematic "

- "iphone wallpaper, conceptual art colorful design, splash of colors, racing car drifting, ultra fine detailed art "

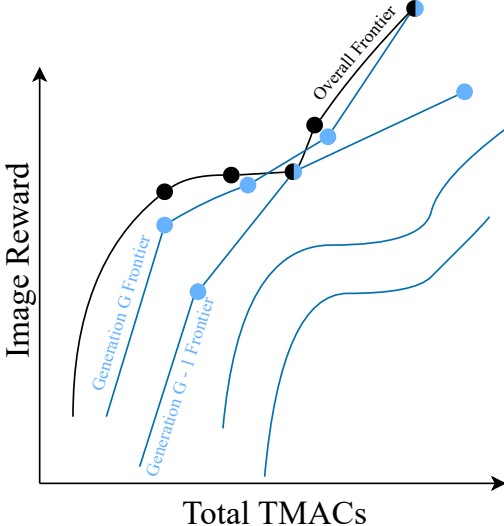

Figure 21: Illustrative example of per-generation and overall Pareto frontiers in ECAD. Points represent candidate schedules, with lines interpolated between them for visualization. Half-colored points lie on both the generational and overall frontiers. In this example, the frontier from generation $G$ appears to exceed the overall frontier, highlighting interpolation 'artifacts' that can occur between discrete candidate solutions.

### A.16 CLARIFYING FRONTIER VISUALIZATIONS

Several frontier plots–such as Figures 16, 17, and 19–show both the Pareto frontier of individual generations (typically shown in color) and the overall frontier aggregated across all generations (typically in black). At first glance, it may seem that a generational frontier occasionally surpasses the overall frontier. This apparent contradiction arises from interpolation between discrete candidate schedules. As illustrated in Figure 21, the frontier from generation $G$ appears to extend beyond the overall frontier. However, the aggregated frontier integrates more finely sampled points, including high-performing candidates from earlier generations (e.g., generation $G-1$), which are not always aligned with the interpolated curves of later generations. The overall frontier, therefore, forms a tighter envelope of all known Pareto-optimal schedules, even if it may visually appear to be exceeded due to interpolation artifacts.

### A.17 LLM USAGE

We utilized LLMs to proofread, check grammar, and suggest revisions during the writing of this manuscript.

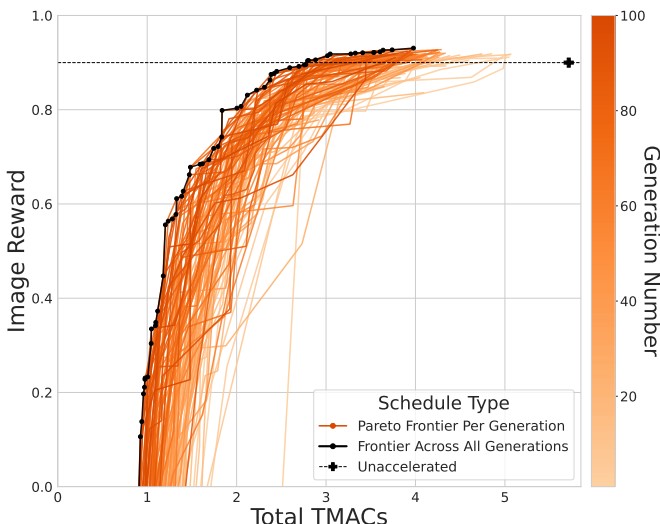

Figure 16: ECAD optimization progress and final Pareto frontier using a reduced population size of 24 (compared to the default of 72), with 100 prompts and 10 images per prompt. The resulting frontiers are noisier and exhibit slower convergence.

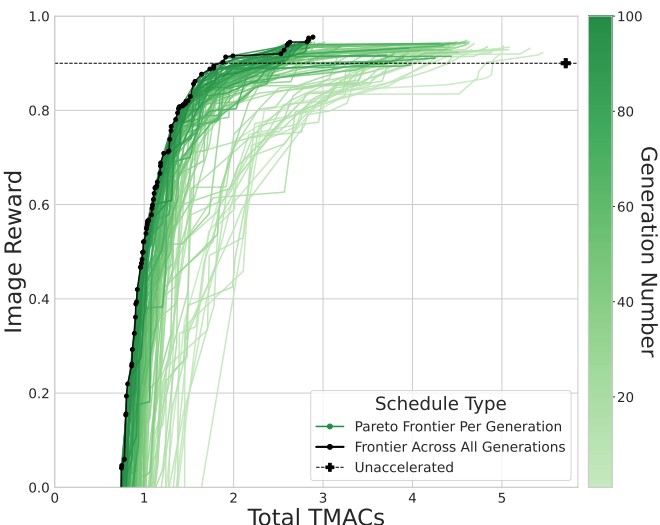

Figure 17: ECAD optimization progress and final Pareto frontier using only 3 images per prompt (default is 10), with 100 prompts and a population size of 72. This configuration demonstrates stable convergence and achieves stronger overall performance.

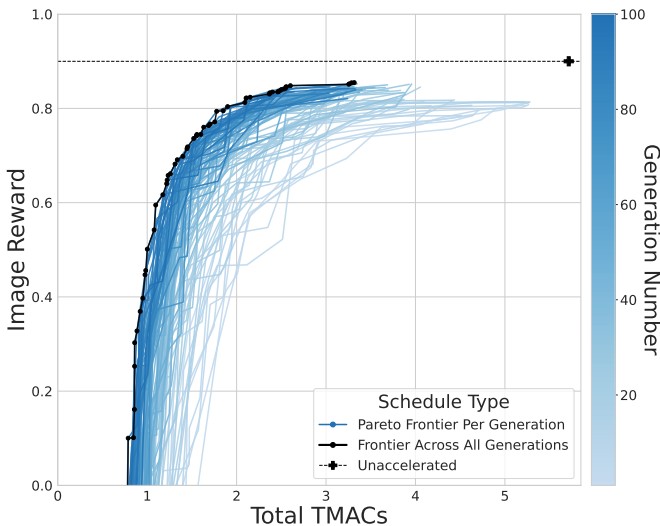

Figure 19: ECAD optimization progress and final Pareto frontier using only 33 prompts (a random subset of the default 100), with 10 images per prompt and population size 72. Although convergence is relatively smooth, the final frontier is constrained by the reduced prompt diversity.

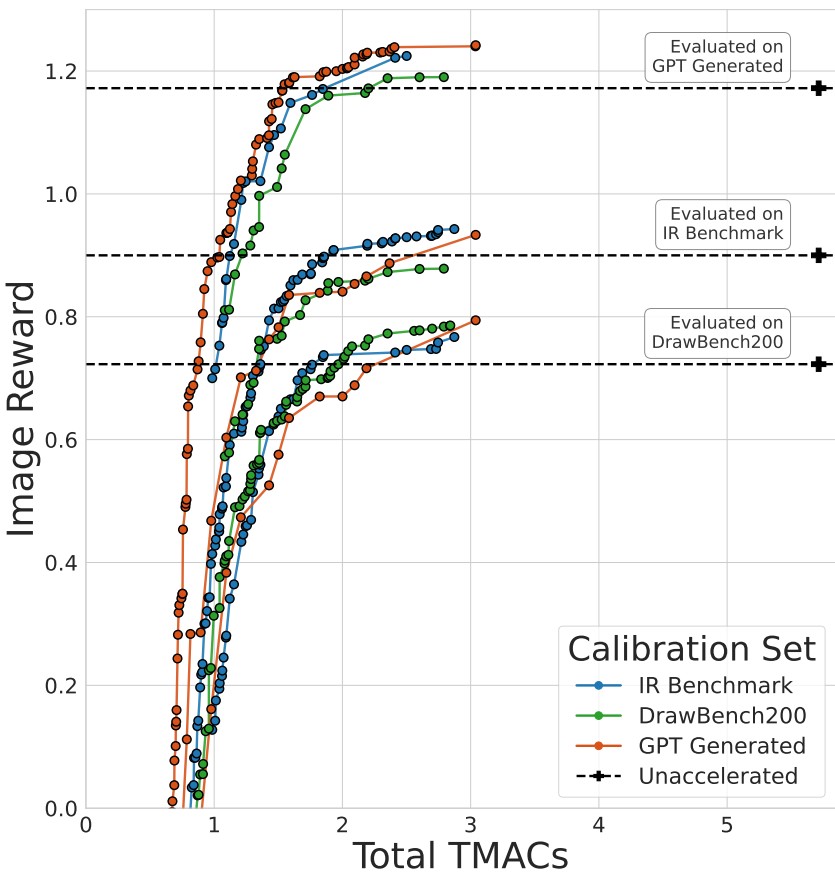

Figure 20: ECAD calibration prompt set ablation. We show performance change when using the human-curated DrawBench200 prompts set and the ChatGPT-generated prompt set for calibration instead of the Image Reward set. Performance is measured in Image Reward (IR) on all combinations of each prompt as used for calibration and evaluation.

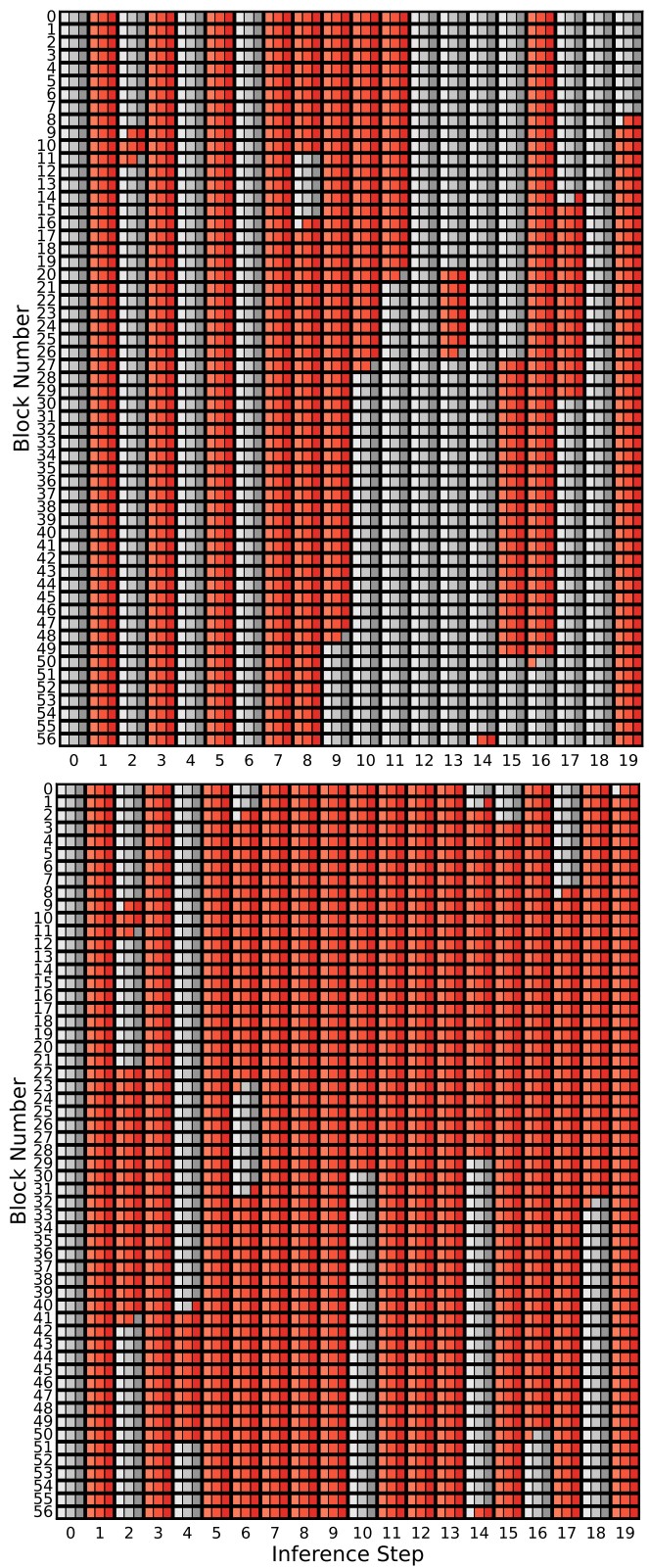

Figure 22: ECAD schedules "slow" (**top**) and "fastest" (**bottom**) for FLUX.1-dev from Table 4 and Table 1 respectively. Despite being almost 200 generations apart, both schedules share similar structures for the first 5 steps, particularly at step 2 for blocks 9 through 12.

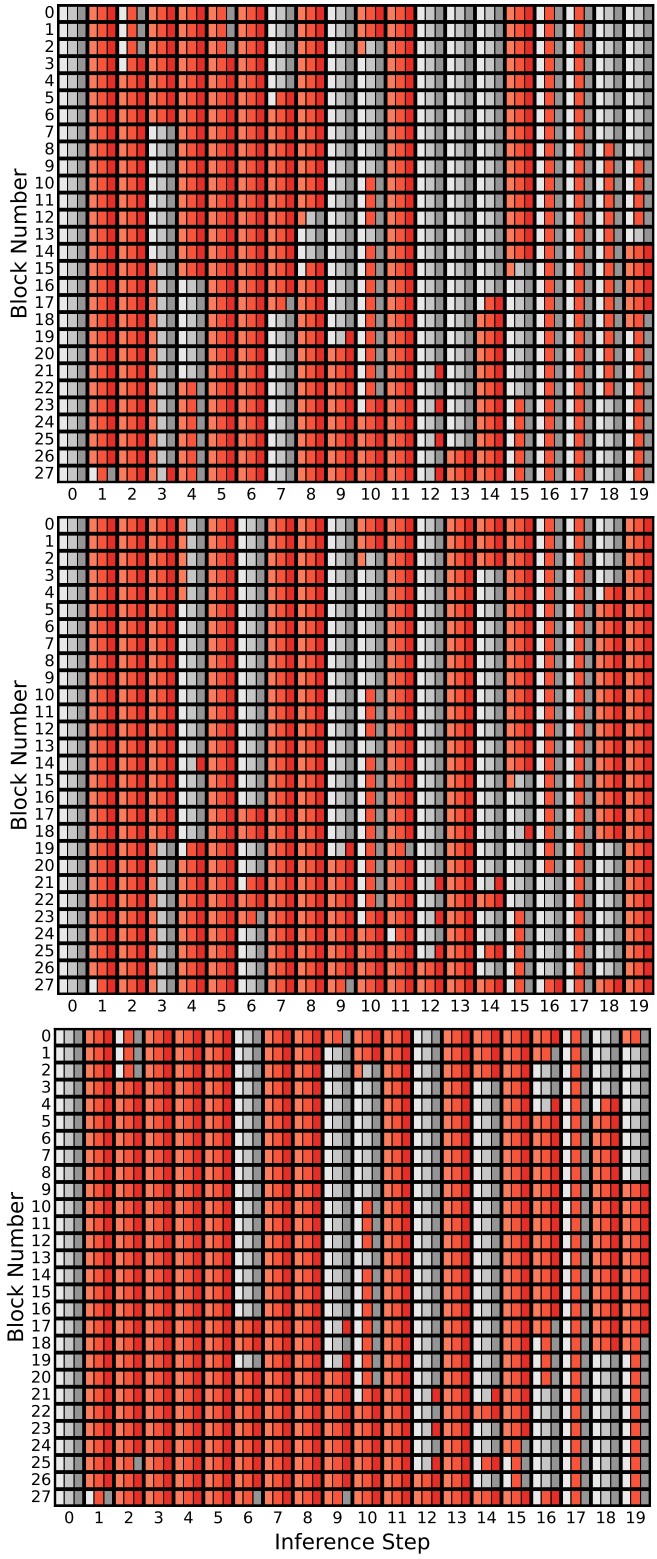

Figure 23: Highest-TMACs schedules from generation 50 (**top**), 200 (**center**), and 400 (**bottom**) during PixArt-$\alpha$ ECAD optimization. While steps between 8 and 15 remain somewhat similar in structure, early and late steps change more.

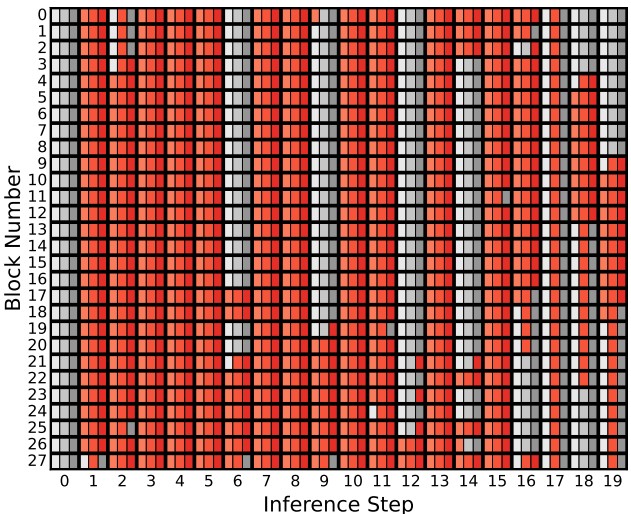

Figure 24: ECAD schedule for PixArt-$\Sigma$ "fast" from Table 1. Initial DiT blocks in steps 6, 9, and 12 are more important to recompute than the final blocks. Cross-attention has less of an impact than the other components in the final three steps, with it as the only component cached in step 17.

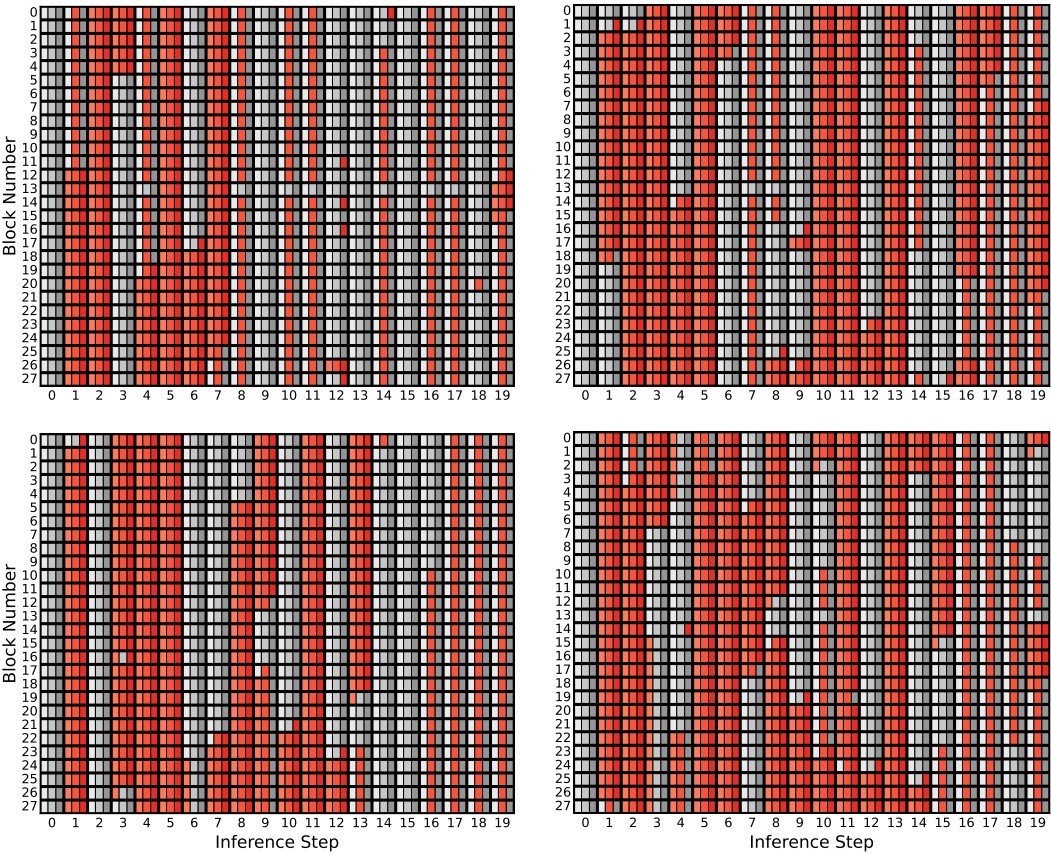

Figure 25: Highest-TMACs schedules after 100 generations for PixArt-$\alpha$ under different hyperparameter ablations: (top-left) reduced population size; (top-right) fewer images per prompt; (bottom-left) fewer prompts; (bottom-right) baseline configuration. All configurations realize the cacheability of cross attention for steps where other components cannot safely be cached.

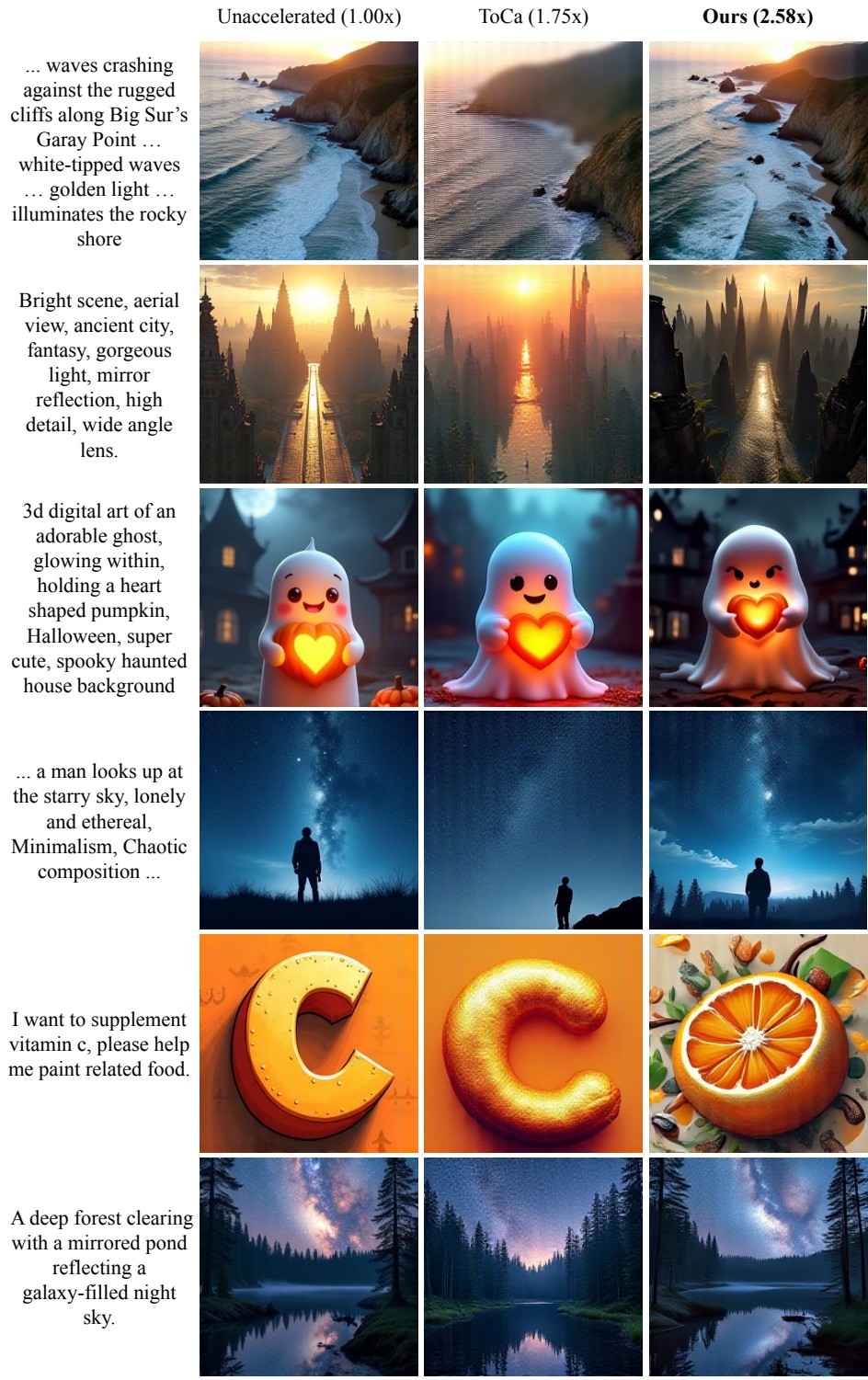

Figure 26: FLUX.1-dev $256 \times 256$ qualitative comparisons. Displayed left-to-right are generations from the uncached baseline, ToCa ($\mathcal{N} = 5, \mathcal{R} = 90\%$; 1.75x speedup), and our "fast" ECAD schedule (Table 1; 2.58x speedup). ECAD consistently yields sharper images with improved prompt adherence.

ToCa (2.47x)

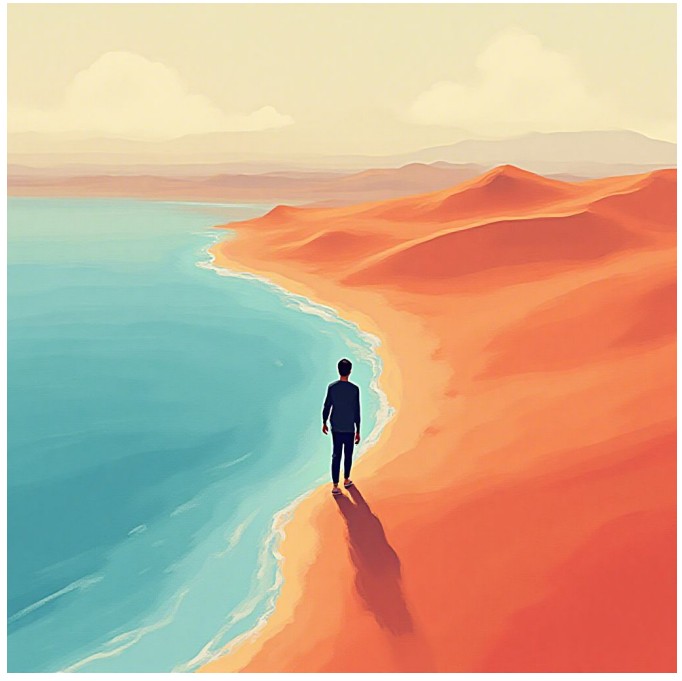

A person standing on the desert, desert waves ... half red, half blue, abstract image of sand ... top view, clear style, precision art

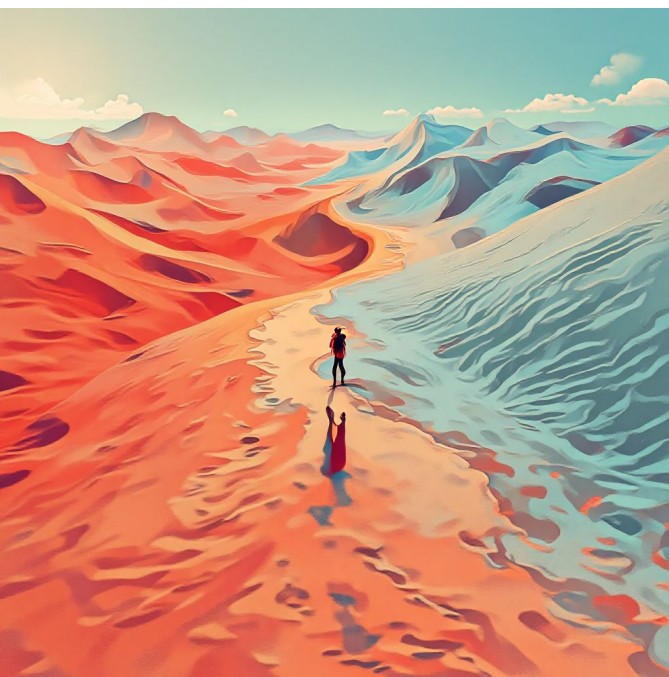

**Ours (2.63x)**

Figure 27: FLUX.1-dev $1024 \times 1024$ qualitative comparisons. Outputs, top-to-bottom, are ToCa ($\mathcal{N} = 4, \mathcal{R} = 90\%$; 2.47x speedup), and our "fast" ECAD schedule (as shown in Table 4; 2.63x speedup). Our method yields greater visual complexity with stronger prompt-alignment, despite higher acceleration.

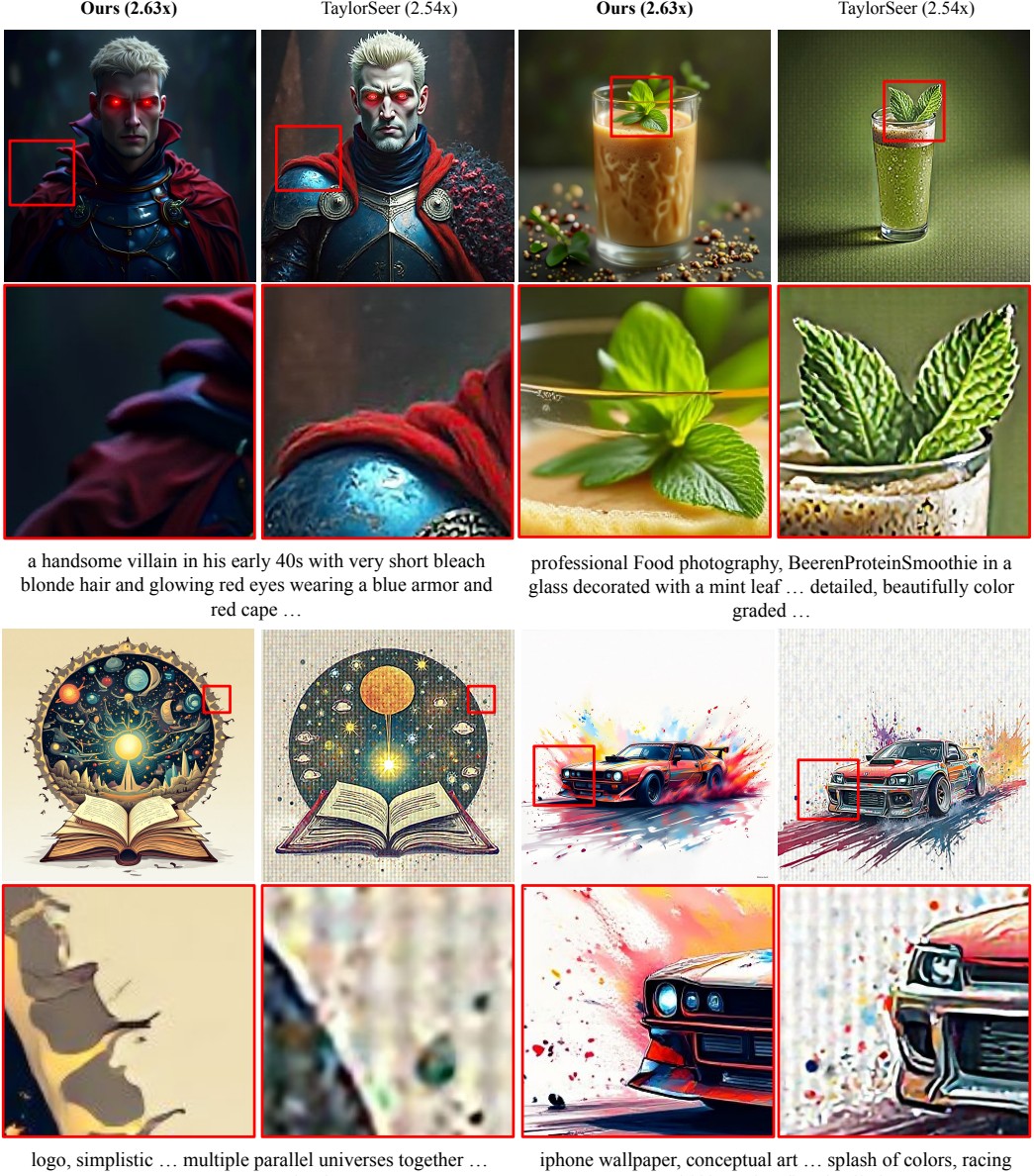

Figure 28: FLUX.1-dev $1024 \times 1024$ further qualitative comparisons. Outputs, left-to-right, are our "fast" ECAD schedule (as shown in Table 4; 2.63x speedup), and TaylorSeer ($\mathcal{N} = 5, \mathcal{O} = 2$; 2.54x speedup). Prompts, from the MJHQ-30K set, and are shown without omission in Appendix A.15. TaylorSeer's method leads to visible patches on solid backgrounds, lower resolution, and color distortion.

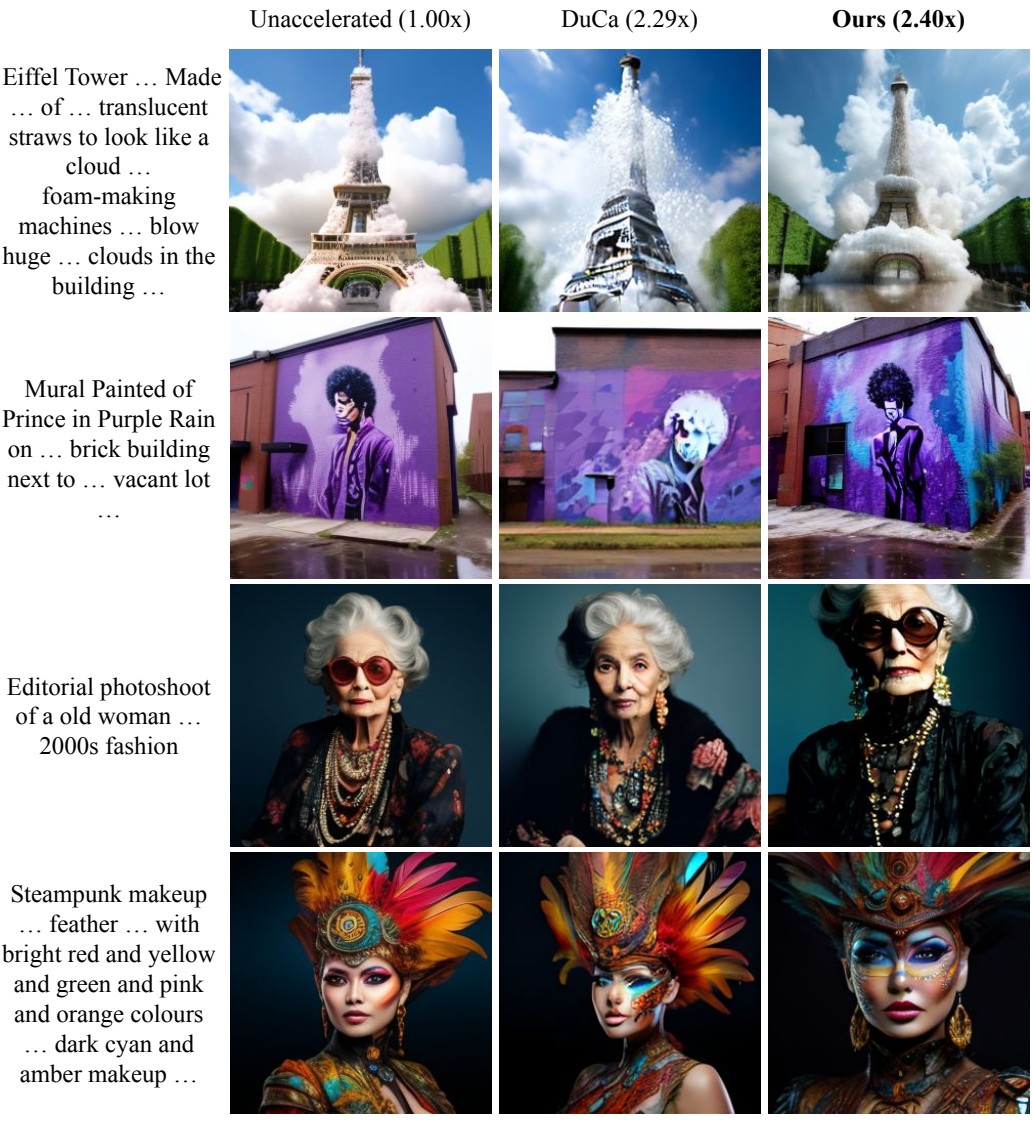

Figure 29: PixArt-$\alpha$ 256$\times$256 further qualitative comparisons. Outputs, left-to-right, are the unaccelerated baseline, DuCa ($\mathcal{N} = 3, \mathcal{R} = 60\%$; 2.29x speedup), and our "faster" ECAD schedule (as shown in Table 1; 2.40x speedup). Our method introduces fewer artifacts and distortions.

