# OpenReview forum: "Evolutionary Caching to Accelerate Your Off-the-Shelf Diffusion Model"
_ICLR.cc/2026/Conference — ICLR 2026 Poster_

### Official Review · Reviewer_NeEf · 2025-10-16

**Soundness:** 3
**Presentation:** 3
**Contribution:** 3
**Rating:** 6
**Confidence:** 4

**Summary:**

This paper proposes a training-free accelerated sampling algorithm for diffusion transformers. The method achieves its speedup by caching and reusing features across multiple inference steps, thereby avoiding redundant computations within the model.

The core innovation of this work is the introduction of the Pareto frontier to construct efficient, model-specific caching schedules. By leveraging evolutionary algorithms to search for this frontier, the authors identify optimal schedules that achieve a favorable trade-off between inference speed and generation quality, using only a minimal set of calibration prompts.

The authors evaluate their method, ECAD, on state-of-the-art models such as PixArt-α, PixArt-Σ, and FLUX-1.dev. Their experiments are conducted across diverse benchmarks (COCO, MJHQ-30k, and PartiPrompts) and assessed with multiple metrics (FID, CLIP Score, and Image Reward), demonstrating consistent performance improvements over previous approaches.

**Strengths:**

1. This paper presents a highly novel approach to the cache-based accelerated sampling problem in diffusion models by leveraging concepts from traditional machine learning. I find the most compelling contribution to be the elegant design of the cache schedule and, in particular, the use of an evolutionary algorithm to efficiently search for the optimal schedule. This represents a creative and effective way to frame and solve the problem. 💡

2. The experimental validation in this work is exceptionally thorough and convincing.

- Comprehensive Evaluation: The authors employ a suite of evaluation metrics, including FID, CLIP Score, and ImageReward, ensuring a multi-faceted assessment of generation quality. Their method is rigorously tested on a diverse range of powerful, state-of-the-art models like PixArt-α, PixArt-Σ, and FLUX-1.dev.

- Insightful Ablations and Visualizations: The paper is further strengthened by thorough ablation studies and clear visualizations. The ablation on the number of evolutionary generations, for instance, provides a clear insight into the trade-off between quality (FID) and efficiency (acceleration ratio), showing how FID first decreases before increasing while the speedup consistently grows. Furthermore, the visualizations of the discovered cache schedules and the points on the Pareto frontiers provide strong, intuitive evidence for the effectiveness of the proposed search strategy.

Overall, the rigorous and well-executed experiments strongly substantiate the claims made in the paper.

**Weaknesses:**

I find several points in the paper confusing, primarily as follows:

1. On the Behavior of Optimization Objectives (ImageReward vs. MACs). I have a question regarding the behavior of the evaluation metrics during the evolutionary search. The paper's objectives are to maximize ImageReward (quality) and minimize MACs (computational cost). I understand how the NSGA-II algorithm operates to find a Pareto frontier for this multi-objective problem. However, the results show that as the number of generations increases, the MACs consistently decrease, while ImageReward does not exhibit a monotonic improvement; instead, it appears to fluctuate. Could the authors clarify this dynamic? My understanding is that while reducing MACs can be achieved by simply caching more, this aggressive caching might negatively impact image quality. Is the fluctuation in ImageReward a result of the algorithm exploring this trade-off, where some generations might sacrifice quality for significant computational savings before finding a better balance? A brief explanation of this trade-off during the search process would be beneficial.

2. On the Representation of the Cache Schedule (S). I know $S \in \{0,1\}^{N,B,C}$, but I don't seem to see $C$ in the visualization. I'd like to understand what role $C$ actually plays.

3. I observe two implicit but critical constraints in the cache schedules: Initial Step Constraint:

- The first inference step (n=0) must always be a full recomputation, as there is no prior cache to reuse. This is reflected in the visualizations.

- Intra-Step Dependency: Within a single inference step n, if a block b is set to recompute, all subsequent blocks (b+1,b+2,…) in that same step must also recompute. A reuse decision for a later block would be illogical, as its input depends on the output of the recomputed block b, invalidating the stored cache.

My question is: how are these logical constraints enforced during the evolutionary search, specifically within the Crossover and Mutation operations? A random mutation or crossover could easily generate an invalid schedule that violates these rules. Do the authors employ constraint-aware genetic operators, or is there a "repair" function that corrects invalid schedules after they are generated? Detailing this mechanism would strengthen the paper's description of the algorithm.

**Questions:**

No

---

> ### Author Response · Authors · 2025-11-21
> **Response to Reviewer NeEf's Feedback**
>
> We thank the reviewer for their encouraging feedback and pleased that the reviewer found the use of an evolutionary algorithm to be a 'creative and effective' solution. We address the reviewer's specfic questions below:
>
> **Behavior of Optimization Objectives**
>
> We would like to clarify the dynamics of the multi-objective optimization process. Unlike single-objective optimization, where a loss function monotonically decreases, NSGA-II seeks a Pareto frontier—a set of solutions where no solution is dominated by another (i.e., better in *both* quality and speed). As such, a schedule that maintains ImageReward score but lowers its TMACs is considered 'Pareto-dominant'. The goal is not necessarily to increase ImageReward above the unaccelerated baseline (though this sometimes happens), but to maintain that high quality while aggressively minimizing MACs.
>
> The fluctuation in ImageReward is indeed a result of the algorithm exploring the trade-off space. As such, some generations may perform worse than the prior generation on certain points along the Pareto frontier.
>
> **Cache Schedule**
> We clarify the definition of $C$ and its visualization:
> $C$ represents the number of cacheable components within each DiT Block. For PixArt-$\alpha$ and PixArt-$\Sigma$, $C=3$, corresponding to the components: self-attention, cross-attention, and feedforward2. Similarly, FLUX.1-dev utilizes $C=3$ cacheable components (detailed in Appendix A.2).
>
> In our heatmaps (e.g., Figures 5, 18, 19), these components are represented as the distinct cells with 3 different shades within a specific (Inference Step, Block Number) pair. A colored cell indicates a cached component, while a grey cell indicates a recomputed component.
>
> **Implicit Constraints and Schedule Validity**
>
> We appreciate the scrutiny regarding schedule validity.
>
> 1. We agree that the first step must always be a full recomputation, and enforce this by treating it as a hard constraint within the optimization problem. Our implementation explicitly forces the first step to be "recompute" during the initialization, crossover, and mutation operations. If a random mutation attempts to set a component at $t=0$ to "cache," it is immediately repaired to ensure validity before evaluation. We appreciate this feedback and will make this revision in the manuscript. **Edit: We have updated this in the manuscript Section A.7, highlighted in blue.**
> 2.  Regarding the dependency between blocks $b$ and $b+1$, we would like to clarify that block $b$ does not need to recompute for block $b+1$ to function. This is due to the residual connections inherent in the Transformer architecture. As illustrated in Figure 2 and Equation 3 in our paper, the output of each component, and each block, is added to the residual stream.
> $$x_{out} = x_{in} + f(x_{in})$$
> When we cache a component, we approximate $f(x_{in})$ using the stored feature from the previous timestep, $M_{cache}$, rather than computing f on the current input as follows:
> $$x_{out} = x_{in} + M_{cache}$$
> Because the residual connection preserves the valid state of the token sequence x, block b+1 receives a valid input tensor regardless of whether block b recomputed its function or fetched it from the cache, and each block and component can be computed independently. While we are unable to provide links, Anthropic's "A Mathematical Framework for Transformer Circuits" provides some fantastic diagrams and mathematical derivations regarding this topic.

---

### Official Review · Reviewer_NiYD · 2025-10-30

**Soundness:** 3
**Presentation:** 3
**Contribution:** 3
**Rating:** 6
**Confidence:** 5

**Summary:**

This paper introduces Evolutionary Caching to Accelerate Diffusion models (ECAD), a genetic algorithm-based framework designed to optimize caching schedules for diffusion models, significantly improving inference speed without compromising image quality. Unlike heuristic-based methods, ECAD formulates caching as a Pareto optimization problem, enabling fine-grained trade-offs between speed and quality. The method is scalable, generalizable across model architectures, and requires no modifications to network parameters. Experiments on PixArt-α, PixArt-Σ, and FLUX-1.dev demonstrate state-of-the-art performance, achieving up to 3.37x speedups while maintaining or improving image quality metrics like FID and Image Reward. ECAD also exhibits strong generalization to unseen resolutions and model variants.

**Strengths:**

+ The proposed evolutIionary caching framework is novel and effective.

**Weaknesses:**

- The authors emphasize in the introduction that "there are no restrictions on batch size." However, it is unclear whether they measure generation performance per image and acceleration ratio under different batch sizes. Doing so would provide strong evidence to support this claim.

- For the strongest model, FLUX, it would be more compelling to evaluate its performance on more comprehensive benchmarks, such as GenEval and DPG-Bench, rather than relying solely on toy datasets.

**Questions:**

see weakness

---

> ### Author Response · Authors · 2025-11-21
> **Response to Reviewer NiYD's Feedback**
>
> We thank the reviewer for their positive assessment and constructive feedback. We are pleased that they found our proposed method both "novel" and "effective". We address the specific questions below.
>
> **Response to Question 1: Clarification on Batch Size Constraints**
>
> We apologize for the confusion regarding the "no restrictions on batch size" claim. This statement refers to the optimization phase, not the inference phase. Unlike gradient-based acceleration methods (e.g., distillation) which require storing optimizer states and gradients (restricting batch size due to VRAM overhead), ECAD is gradient-free. This enables efficient optimization on even consumer hardware.
>
> For inference, we simulated realistic production throughput by utilizing the maximum memory-fitting batch size (B) for all latecy timing and equality evaluations, as detailed in Appendix Table 12.
>
> **Response to Question 2: DPG Bench & GenEval for FLUX**
>
> We agree that evaluating FLUX on more comprehensive benchmarks strengthens our claims. We are currently running our "Fast" and "Fastest" schedules on GenEval (553 prompts) and DPG-Bench (1065 prompts) to demonstrate robustness and will update this response with the resulting metrics shortly.

---

> > ### Author Response · Authors · 2025-11-24
> > **Follow Up to Response to Reviewer NiYD's Feedback**
> >
> > We thank the reviewer for their patience while we prepared results on GenEval and DPG Bench. We have revised the manuscript (changes highlighted in blue) and included these results in Section A.9. We compare to the same prior works from the main paper's Table 1. We find our "Fast" schedule outperforms all such works in image quality, while at higher acceleration. Additionally, the "Fastest" schedule only results in slight to modest quality loss at 3.37x acceleration.
> >
> > We have reproduced the table below, with full details in the revised manuscript Section A.9.
> >
> > **Table: R2: FLUX-1.dev Performance on GenEval and DPG Bench.**
> > We compare our and other methods from Table 1 on the GenEval and DPG Bench benchmarks using FLUX-1.dev (20 steps, 256×256). Our method does not impact GenEval Overall score at 2.58× acceleration, while other methods result in 2% to 22% quality decrease for lower acceleration. Our method achieves the highest speedups while even slightly improving the DPG Bench score, whereas other aggressive caching strategies degrade performance.
> >
> > | Caching    | Setting     | TMACs↓     | ms / img↓ (speedup↑) | GenEval Overall Score | GenEval Overall % Decrease | DPG Bench Score | DPG Bench % Decrease |
> > | ---------- | ----------- | ---------- | -------------------- | --------------------- | -------------------------- | --------------- | -------------------- |
> > | None       |             | 198.69     | 2620.09 (1.00x)      | 0.5842                | --                         | 22.7058         | --                   |
> > | ToCa       | N=4, R=90%  | **42.96*** | 1576.97 (1.66x)*     | 0.5517                | 5.56%                      | 22.8215         | -0.51%               |
> > | DiCache    |             | 62.23      | 1161.86 (2.26x)      | 0.5699                | 2.45%                      | 22.6946         | 0.05%                |
> > | TaylorSeer | N=5, O=2    | 59.88*     | 1028.66 (2.55x)*     | 0.4531                | 22.44%                     | 22.4695         | 1.04%                |
> > | TaylorSeer | N=6, O=1    | 49.97*     | 865.97 (3.03x)*      | 0.3399                | 41.81%                     | 21.6869         | 4.49%                |
> > | **Ours**   | **Fast**    | 63.02      | 1016.59 (2.58x)      | **0.5892**            | **-0.86%**                 | **22.8364**     | **-0.58%**           |
> > | **Ours**   | **Fastest** | 43.60      | **778.17 (3.37x)**   | 0.5258                | 10.00%                     | 23.5098         | -3.54%               |

---

> > > ### Comment · Reviewer_NiYD · 2025-11-28
> > >
> > > Thank you for your detailed response. Please add them to the Appendix to facilitate future research.

---

### Official Review · Reviewer_X7Lz · 2025-11-01

**Soundness:** 3
**Presentation:** 3
**Contribution:** 3
**Rating:** 6
**Confidence:** 2

**Summary:**

This paper introduces ECAD, a method that uses genetic algorithms to learn optimal caching schedules for accelerating Diffusion Transformers. The authors demonstrate speedups on models like PixArt and Flux while preserving or even improving image quality across multiple benchmarks.

**Strengths:**

- framing diffusion caching as a multi-objective Pareto optimization problem to balance quality and efficiency is novel.
- the proposed method is lightweight, and can adapt across prompts, model variants, and resolutions.
- strong empirical performance that achieves state-of-the-art inference acceleration with competitive image quality across several models.
- the paper is well-written and the comprehensive ablation studies in the appendix provide supports for the design choices.

**Weaknesses:**

- the performance is dependent on the choice of the quality metric used during optimization, and the caching schedules may not generalize perfectly to all possible evaluation metrics.
- the experiment setup (e.g., 100 prompts) may not be representative of complex prompts from real-world applications.

**Questions:**

- what's the generalizability of a schedule optimized for a certain reward, e.g. Image Reward, when evaluated with a different metric?
- how stable are the Pareto frontiers across random seeds and initializations of population, versus being seeded with heuristic schedules?
- can you analyze the selected prompt in terms of how representative they are of real-world use, and how sensitive is the performance of the proposed method towards prompts with different length, granularity, etc?

---

> ### Author Response · Authors · 2025-11-21
> **Response to Reviewer X7Lz's Feedback**
>
> Thank you for the thoughtful and constructive review. We appreciate the reviewer’s recognition of ECAD’s novelty, empirical strength, and clarity, and we address each of the raised questions and concerns below.
>
> **Response to Question 1 and Weakness 1: Dependency on Quality Metric**
>
> Our method is not dependent on Image Reward. When we calibrate using only ImageReward, Table 1 shows that we achieve good performance for completely different metrics (CLIP score, FID), as well as for the same metric but with more challenging prompts (PartiPrompts). Sections A.3 and Table 6 further address the issue: we apply ECAD with a completely different metric (weighted combination of CLIP score and CLIP IQA) and still outperform ToCa (in terms of speed, and 3 out of 4 quality metrics).
>
>
> **Response to Weakness 2: The Experiment Setup**
>
> While we only need 100 prompts for calibration, our actual experiments test tens of thousands of prompts along diverse axes. MJHQ has 30,000 prompts to test aesthetic quality. COCO has 30,000 prompts which correspond to actual real-world descriptions of scenes with many diverse objects. PartiPrompts has 1,632 prompts which test animals, world knowledge, outdoor scenes, vehicles, food, and more. We perform well on all of these, providing evidence that our method finds schedules that can accelerate diffusion models while still achieving good results on diverse, real-world prompts.
>
> **Response to Question 2: Initial Seeds**
>
> Fully random seeds are not sufficient for our method, since the genetic algorithm needs a good distribution of schedules in terms of TMACs. We can resolve this by making sure the random schedules are uniform in terms of computation, or by using a heuristic. In our case, we find it simplest to simply build on foundational worlds in the area (FoRA) and use such schedules to design initial seeds.  This is flexible and easy to apply to calibrate for any given model.
>
> **Response to Question 3: Sensitivity Towards Prompts**
>
> We analyze details of prompt set selection in Appendix A.4, with results in Table 7 and Table 8. In terms of how representative they are of real world use, we evaluate on diverse datasets (MJHQ, COCO, PartiPrompts) all with different prompt dynamics to try to approximate how well our schedules align with real world prompts. We find the performance is overall quite robust. We get good performance for datasets focused on objects (COCO) and animals/people/vehicles/food/"world knowledge" (PartiPrompts) even when we calibrate with ChatGPT-written prompts focused only on landscapes (Table 8). ECAD finds robust schedules that shorten diffusion inference without compromising generalizability, even when the calibration prompts focus on a narrow topic.

---

> > ### Author Response · Authors · 2025-11-24
> > **Follow Up to Response to Reviewer X7Lz's Feedback**
> >
> > To better address your questions and concerns, we include further details below. We have revised the manuscript (PDF changes highlighted in blue) to address Question 2 in Section A.8.
> > We compare our heuristic initialization to two forms of random initialization - a random sample of binary caching tensors, which performs poorly, and a random sample of computational complexity (as measured by TMACs). Figure 11 and Table 12 show the randomly sampling computational complexity performs very well, but heuristics remain preferable. Please see further details in the manuscript.
> >
> > Second, to directly address Question 3 and quantify the impact of prompt length/granularity, we have added a new calibration-set ablation, summarized in **Table R1** below.
> >
> > **Table R1: Calibration Prompt Set Size Ablation.** We include two additional calibration prompt sets for reference, in addition to the ImageReward Benchmark baseline and Painted Landscapes ablation from Section A.4. The schedule with the highest TMACs is shown after 100 generations, except for "5 Word Prompts (Faster)", which is selected to provide an additional point of reference.
> > | Calibration Set                | ms / img↓(speedup↑) | Calib. Set IR↑ | PP IR↑ | COCO FID↓ | COCO CLIP↑ | MJHQ FID↓ | MJHQ CLIP↑ |
> > | ------------------------------ | ------------------- | -------------- | ------ | --------- | ---------- | --------- | ---------- |
> > | ImageReward Benchmark          | 100.68   (1.65x)    | 0.94           | **1.0**| 21.4      | 31.48      | 8.18      | **32.88**  |
> > | Painted Landscapes             | 95.41    (1.74x)    | 1.2            | 0.97   | 20.82     | **31.58**  | 8.55      | 32.85      |
> > | 10 Prompts                     | 97.87    (1.69x)    | **1.31**       | 0.94   | 25.27     | 31.51      | 10.02     | 32.84      |
> > | 5 Word Prompts (Highest TMACs) | 109.18   (1.52x)    | 1.01           | 0.98   | **20.05** | 31.46      | **7.42**  | 32.82      |
> > | 5 Word Prompts (Faster)        | **95.27  (1.74x)**  | 1.0            | 1.0    | 21.68     | 31.36      | 8.76      | 32.76      |
> >
> > Table R1 shows performance on our evaluation sets (PartiPrompts, MJHQ-30K, COCO-30K) to remain relatively stable, despite the varried calibtration prompt set selection. The ablation using a set of 100 five-word prompts achieves better performance than all prior methods (see Main Paper Table 1) at slightly lower acceleration (1.74x). Combined with our strong results using a set of 100 prompts of only painted landscapes, this shows ECAD to be robust against short/coarse and task-specific prompt sets.
> > We have additionally included an ablation using only 10 prompts, which results in lowered performance, similar to the results shown in Section A.4 with 33 prompts. Thus, the contents of the prompt set matters less, while the quantity matters more.
> >
> > Further details:
> >
> > We generate a compact calibration set of 10 prompts using ChatGPT (Web, 5.1 Instant) with the following instruction:
> >    > Generate 10 prompts for benchmarking image generation models (such as PixArt Alpha, Stable Diffusion, etc.). The prompts should be diverse and cover a wide range of styles, including photorealism, painting, anime, pixel art, and more. Each prompt should be crafted to evaluate aspects like aesthetics, compositional accuracy, text rendering, and subject diversity, in order to comprehensively test model quality.
> >
> > Since we fix the number of images generated per schedule evaluation to 1000, this ablation generates 100 images per prompt. All other hyperparameters are held constant relative to the baseline setup. We report the performance of the schedule on the Pareto frontier after 100 generations with the highest TMACs (row "10 Prompts (GPT)" in the table).
> >
> > Sample prompts:
> > > A hyper-realistic close-up portrait of an elderly woman with deep wrinkles, silver hair strands illuminated by soft window light, and subtle reflections in her eyes; shallow depth of field and natural skin texture.
> > > A bustling open-air market at dusk with dozens of vendors, colorful produce, hanging lanterns, and people interacting naturally; accurate perspective, occlusion, and coherent shadows.
> >
> >
> > We also construct a set of 100 very short prompts (each with at most 5 words) with prompt:
> >    > Generate 100 prompts for benchmarking image generation models (such as PixArt Alpha, Stable Diffusion, etc.). The prompts should be brief (e.g. not granular), with no more than 5 words per prompt.
> >
> > Aside from the prompt text itself, all settings are identical to the baseline ImageReward Benchmark (same number of images, generations, optimization hyperparameters). Because the largest schedule discovered under this calibration achieves only a 1.52× speedup, we report both the highest-TMACs schedule (“5 Word Prompts (GPT) - largest”) and a faster schedule on the frontier (“5 Word Prompts (GPT) - fast”) to provide an additional point of reference.
> >
> > Sample prompts:
> > > Neon city at night
> > > Ancient desert ruins
> > > Snowy mountain village

---

### Official Review · Reviewer_Mtyb · 2025-11-01

**Soundness:** 3
**Presentation:** 3
**Contribution:** 3
**Rating:** 6
**Confidence:** 4

**Summary:**

This paper, “Evolutionary Caching to Accelerate Your Off-the-Shelf Diffusion Model (ECAD)”, addresses the problem of slow and computationally intensive inference in diffusion-based generative models, especially Diffusion Transformers (DiTs). Traditional caching methods reuse previously computed features during sequential inference steps to reduce redundant computation, but existing approaches rely heavily on rigid heuristics, hand-tuned hyperparameters, and limited trade-off flexibility. ECAD reframes feature caching as a multi-objective optimization problem over image quality and computational efficiency, solved via a genetic algorithm (specifically, NSGA-II). The algorithm evolves caching schedules—which specify what components (self-attention, cross-attention, feedforward outputs) to cache, and when—to form a Pareto frontier of quality–latency trade-offs. ECAD is model-agnostic, training-free, and learns these schedules using only a small set of “calibration prompts” and a quality metric (e.g., Image Reward). Experiments on three major diffusion models (PixArt-α, PixArt-Σ, FLUX-1.dev) show that ECAD provides higher-quality outputs at faster inference speeds compared to previous caching baselines such as FORA, ToCa, DuCa, and TaylorSeer. The method is also shown to generalize across resolutions (e.g., 256×256 to 1024×1024) and related model architectures without retraining.

**Strengths:**

1. **Novel framing and methodological clarity.**
    The paper introduces a principled optimization perspective (Pareto frontier with NSGA-II) that replaces heuristic-based caching rules. This reframing is conceptually elegant and well-motivated.

2. **Training-free and model-agnostic.**
    ECAD operates without any weight updates or retraining, making it practical for various diffusion models—especially for users without access to large compute resources.

3. **Comprehensive experimental evaluation.**
    Evaluations cover multiple high-profile models (PixArt and FLUX families), a range of datasets (COCO, MJHQ-30K, PartiPrompts), and metrics (FID, CLIP, Image Reward), showing consistent improvements.on prompts, and different reward functions

4. **Clear writing and structured presentation.**
    The paper is well-organized, readable, and includes detailed algorithmic pseudocode and visualizations.

**Weaknesses:**

1. **Compute inefficiency during optimization.**
    Although no network training is required, ECAD’s optimization process (hundreds of generations, thousands of images per prompt) might still be computationally expensive. The paper could better quantify this cost relative to real-world savings.

2. **Reliance on automatic quality metrics.**
    The optimization depends on Image Reward or similar automatic metrics, whose correlation to human preference can be imperfect or domain-specific. The paper does not show any human evaluation.

3. **Potential overfitting to calibration prompts.**
    While generalization is discussed, there’s limited theoretical grounding for why a small prompt set suffices to generalize to unseen data.

4. **Reproducibility complexity.**
    The large-scale evolutionary optimization (e.g., 500 generations × 72 candidates) may be difficult for others to reproduce without substantial GPU time.

**Questions:**

1. How computationally heavy is ECAD’s optimization stage in practice (e.g., total GPU-hours)? Is the cost justified by the inference savings for typical users?

2. What are the performance trade-offs if the number of calibration prompts is reduced further (e.g., to 10 or 20)? How stable is ECAD under smaller calibration sets?

3. Could ECAD be combined with other acceleration strategies (e.g., distillation or quantization)? Would the Pareto optimization remain consistent?

---

> ### Author Response · Authors · 2025-11-21
> **Response to Reviewer Mtyb's Feedback**
>
> We thank the reviewer for their time, thoughtful evaluation, and kind words regarding the writing quality. We are encouraged that you found our method novel, "elegant," and "well-motivated." We address your specific questions and concerns below.
>
> **Response to Weakness 1, 4, & Question 1: Cost and Reproducibility**
>
> We appreciate the opportunity to clarify the computational overhead of ECAD. We view the optimization phase as a one-time 'fixed cost' that is amortized over the lifetime of a deployed model.
> ECAD’s optimization occurs only once per model, and can even be transfered between similiar model architectures (Table 3). The output is the lightweight caching schedule that can be shared and reused universally. Consequently, only one person/group needs to perform the optimization and release the set of schedules, benefiting the entire community. For the end-user, the cost is zero; they simply select a pre-optimized schedule that meets their latency needs.
>
> Our work focuses on a core research contribution for algorithmic inference speedup without loss (compared to distillation which accepts significant penalties to FID and CLIP), and further engineering efforts could reduce GPU time. To discover our "Faster" and "Fastest" PixArt-$\alpha$ configurations in 548 generations, it takes only  699 NVIDIA A6000 GPU hours. Additionally, we would like to emphasize that this results in a whole frontier of schedules, not just a single or select few configurations, unlike other methods. Further, discovering the "Fast" schedule is even quicker in 358 generations -- only 470 GPU-hours. Running 100 generations with our default settings results in a schedule with a 16% reduction in MJHQ FID over baseline and takes just 145 hours.
>
> With slight engineering tweaks, we discover the schedule corresponding to Row 3 of Table 10 in 100 generations, **a total of 44 GPU-hours**! This schedule achieves *better* performance than the unaccelerated baseline, with an 11.8% MJHQ FID decrease and 2% increase in PartiPrompts score (both are unseen test sets). For large scale deployments, a service using our 2.58x or even 3.37x FLUX models would recover that GPU optimization debt very quickly, and each additional generation of optimization they do takes fewer and fewer GPU-hours.
>
> We will update the manuscript to including these reporting numbers; thank you for the valuable feedback.
>
> Regarding Weakness 4 specifically, we wish to highlight that ECAD is more accessible than standard training-based acceleration methods. It does not require gradient computation, and as such does not have additional VRAM requirements associated with backpropagation. Additionally, optimization is fully asynchronous and can be distributed across a cluster of heterogeneous, lower-end consumer GPUs, rather than requiring high-end H100/A100 clusters. From our own observation, the generated images also more closely resemble the style and aesthetic properties of the model we accelerate.
>
>
>
> **Response to Weakness 2 & 3: Generalization and Metric Robustness**
>
> While we do not propose a theoretical framework for prompt sufficiency, we feel our extensive empirical results demonstrate that ECAD avoids overfitting to either the specific metrics or the limited calibration sets used (as shown across COCO, MJHQ, PartiPrompts, with FID & CLIP).
>
> We utilize ImageReward because it is trained on human preference data and correlates well with human judgment. However, ECAD is fundamentally metric-agnostic; i.e. as a framework ECAD does not depend on a particular metric. Its asynchronous design supports any reward signal, allowing practitioners to substitute ImageReward with direct human evaluation or other metrics without architectural changes (see Appendix A.3).
>
>
> **Response to Question 2: Stability with Fewer Prompts**
>
> We appreciate the suggestion to probe the lower bounds of our calibration set. We view 100 prompts as a simple lower-bound requirement given the ease of automated generation and have included results in Section A.4 demonstrating strong performance with a set of 100 LLM-generated prompts.
> We agree that testing stability under extreme sparsity provides valuable insight into the method's robustness and are currently conducting an ablation with a 10-prompt set and will update the reviewer with the results shortly.
>
> **Response to Question 3: Combination with Other Acceleration Strategies**
>
> Yes, ECAD is fully compatible with strategies like quantization or distillation. As a black-box optimizer, ECAD simply treats the compressed model as a new baseline. Additionally, as shown in Section 4.3, schedules from the original model can be used to initialize the population for the compressed model, accelerating convergence.

---

> ### Author Response · Authors · 2025-11-24
> **Follow Up to "Initial Response to Reviewer Mtyb's Feedback"**
>
> We thank the reviewer for their patience; we have ran additional experiments and revised the manuscript accordingly.
>
> First, as requested, we have added an explicit accounting of ECAD’s optimization cost in GPU-hours to Section A.10 of the revised paper (changes highlighted in blue). This section now reports end-to-end GPU-hours for discovering our “fast”, “faster”, and “fastest” schedules on PixArt-α.
>
> Second, to directly address your concern about stability under smaller calibration sets and potential overfitting to a limited prompt set (Weakness 3, Question 2), we have added a new calibration-set ablation, summarized in **Table R1** below.
>
> **Table R1: Calibration Prompt Set Size Ablation.** We include two additional calibration prompt sets for reference, in addition to the ImageReward Benchmark baseline and Painted Landscapes ablation from Section A.4. The schedule with the highest TMACs is shown after 100 generations, except for "5 Word Prompts (Faster)", which is selected to provide an additional point of reference.
> | Calibration Set                | ms / img↓(speedup↑) | Calib. Set IR↑ | PP IR↑ | COCO FID↓ | COCO CLIP↑ | MJHQ FID↓ | MJHQ CLIP↑ |
> | ------------------------------ | ------------------- | -------------- | ------ | --------- | ---------- | --------- | ---------- |
> | ImageReward Benchmark          | 100.68   (1.65x)    | 0.94           | **1.0**| 21.4      | 31.48      | 8.18      | **32.88**  |
> | Painted Landscapes             | 95.41    (1.74x)    | 1.2            | 0.97   | 20.82     | **31.58**  | 8.55      | 32.85      |
> | 10 Prompts                     | 97.87    (1.69x)    | **1.31**       | 0.94   | 25.27     | 31.51      | 10.02     | 32.84      |
> | 5 Word Prompts (Highest TMACs) | 109.18   (1.52x)    | 1.01           | 0.98   | **20.05** | 31.46      | **7.42**  | 32.82      |
> | 5 Word Prompts (Faster)        | **95.27  (1.74x)**  | 1.0            | 1.0    | 21.68     | 31.36      | 8.76      | 32.76      |
>
> The ablation using a set of 100 five-word prompts achieves better performance than all prior methods (see Main Paper Table 1) at slightly lower acceleration (1.74x). Combined with our strong results using a set of 100 prompts of only painted landscapes, this shows ECAD to be robust against short/coarse and task-specific prompt sets. As requested, we have included an ablation using only 10 prompts. This results in lowered performance, similar to the results shown in Section A.4 with 33 prompts. Thus, the contents of the prompt set matters less, while the quantity matters more. Please note all ablations are conducted with 1000 images generated per schedule, so these results show that increasing the size of the prompt set while reducing the number of images generated per prompt would result in improved performance for the same compute.
>
> Further details:
>
> We generate a compact calibration set of 10 prompts using ChatGPT (Web, 5.1 Instant) with the following instruction:
> > Generate 10 prompts for benchmarking image generation models (such as PixArt Alpha, Stable Diffusion, etc.). The prompts should be diverse and cover a wide range of styles, including photorealism, painting, anime, pixel art, and more. Each prompt should be crafted to evaluate aspects like aesthetics, compositional accuracy, text rendering, and subject diversity, in order to comprehensively test model quality.
>
> Since we fix the number of images generated per schedule evaluation to 1000, this ablation generates 100 images per prompt. All other hyperparameters are held constant relative to the baseline setup. We report the performance of the schedule on the Pareto frontier after 100 generations with the highest TMACs (row "10 Prompts (GPT)" in the table).
>
> Sample prompts:
> > A hyper-realistic close-up portrait of an elderly woman with deep wrinkles, silver hair strands illuminated by soft window light, and subtle reflections in her eyes; shallow depth of field and natural skin texture.
> > A bustling open-air market at dusk with dozens of vendors, colorful produce, hanging lanterns, and people interacting naturally; accurate perspective, occlusion, and coherent shadows.
>
> We also construct a set of 100 very short prompts (each with at most 5 words) with prompt:
> > Generate 100 prompts for benchmarking image generation models (such as PixArt Alpha, Stable Diffusion, etc.). The prompts should be brief (e.g. not granular), with no more than 5 words per prompt.
>
> Aside from the prompt text itself, all settings are identical to the baseline ImageReward Benchmark calibration (same number of images, generations, and optimization hyperparameters). Because the largest schedule discovered under this calibration achieves only a 1.52× speedup, we report both the highest-TMACs schedule (“5 Word Prompts (GPT) - largest”) and a faster schedule on the frontier (“5 Word Prompts (GPT) - fast”) to provide an additional point of reference.
>
> Sample prompts:
> > Neon city at night
> > Ancient desert ruins
> > Snowy mountain village

---

### Author Response · Authors · 2025-11-30
**Summary for Area Chairs**

## Note to the Area Chairs
We understand this year presents unique challenges and deeply appreciate the Area Chair(s) taking on the unprecedented workload of resolving the rebuttal and discussion phase more directly under these circumstances. Thank you for your time and effort in evaluating our work.

## Highlights from Reviews
ECAD redefines diffusion acceleration by replacing rigid heuristics with a **"conceptually elegant"** (Reviewer Mtyb) evolutionary framework that learns optimal caching schedules. Recognized by reviewers as a **"highly novel approach,"** (Reviewer NeEf) our method treats speed and quality as a Pareto optimization problem, delivering **"state-of-the-art inference acceleration"** (up to 3.37x; Reviewer X7Lz) across PixArt and FLUX models. With **"strong empirical performance"** (Reviewer X7Lz) and **"exceptionally thorough"** (Reviewer NeEf) validation, ECAD offers a **"creative and effective"** (Reviewer NeEf) training-free solution that consistently outperforms existing baselines while maintaining visual fidelity.


## Addressing Questions and Concerns
Across 4 reviews, reviewers raised 11 distinct questions and concerns regarding the experimental scope, algorithmic details, and optimization costs. We resolved 5 of these points through detailed clarification in our rebuttals, addressed the remaining 6 concerns by conducting new experiments and analyses, and made 4 key additions to the manuscript.

## Key Manuscript Changes
All changes are highlighted in blue in the updated PDF. Key manuscript additions:

* **Calibration Set Robustness (Section A.4)**: To address concerns regarding overfitting and prompt selection, we expanded Section A.4 with additional ablations. We tested a minimal set (10 prompts) and a coarse set (100 prompts, <=5 words). This demonstrated ECAD as robust to prompt granularity, achieving state-of-the-art results with simple 5-word prompts, provided the calibration set size remains sufficiently sized.

* **Initialization Analysis (Section A.8)**: We compare heuristic initialization against "True Random" and a "Uniform Random" baseline. We demonstrate that heuristics yield superior convergence and speedups, while "Uniform Random" sampling avoids the diversity collapse observed in naive "True Random" binary sampling strategies.

* **Extended Benchmarks for FLUX (Section A.9):** As requested, we perform evaluation of FLUX on GenEval and DPG-Bench. Our "Fast" schedule achieves 2.58x speedup with slight improvements in GenEval and DPG Bench scores over the unaccelerated baseline, outperforming prior methods like ToCa, TaylorSeer, and DiCache.

* **Computational Cost Analysis (Section A.10):** To address concerns regarding optimization overhead, we explicitly report the GPU-hours required for ECAD and demonstrate that further engineering efforts result in further efficiency gains. We demonstrate that discovering a high-performance 1.65x schedule with an 11.8% reduction in FID over baseline costs 44 A6000 GPU-hours -- a one-time fixed cost amortized over model deployment.

---

### Meta-Review · Area_Chair_VZeb · 2025-12-19

**Summary:**

The reviewers raised concerns about dependency on automatic quality metrics, sensitivity to calibration prompt selection, the need for evaluation on larger benchmark datasets, and some clarification questions about the algorithm design, cache schedule constraints, and experimental setup. Most of the concerns were addressed by the author rebuttal.

**Reviewer Concerns:**

The concerns about sensitivity to calibration prompt selection and the need for evaluation on larger benchmark datasets were addressed by additional experiments. Additional information were provided during rebuttal to address the concerns about dependency on metrics, algorithm design, cache schedule constraints, and experimental setup. Most concerns were addressed.

**Reviewer Scores:**

It's likely that all the reviewers will keep the postive scores.

---

### Decision · Program_Chairs · 2026-01-26

Accept (Poster)